# Uplift Modeling for Target User Attacks on Recommender Systems

## Submission ID: 507

## ABSTRACT

Recommender systems are vulnerable to injective attacks, which inject limited fake users into the platforms to manipulate the exposure of target items to all users. In this work, we identify that conventional injective attackers overlook the fact that each item has its unique potential audience, and meanwhile, the attack difficulty across different users varies. Blindly attacking all users will result in a waste of fake user budgets and inferior attack performance. To address these issues, we focus on an under-explored attack task called target user attacks, aiming at promoting target items to a particular user group. In addition, we formulate the varying attack difficulty as heterogeneous treatment effects through a causal lens and propose an Uplift-guided Budget Allocation (UBA) framework. UBA estimates the treatment effect on each target user and optimizes the allocation of fake user budgets to maximize the attack performance. Theoretical and empirical analysis demonstrates the rationality of treatment effect estimation methods of UBA. By instantiating UBA on multiple attackers, we conduct extensive experiments on three datasets under various settings with different target items, target users, fake user budgets, victim models, and defense models, validating the effectiveness and robustness of UBA.

• **Relevance:** As required by WWW'24 CFP, we state that this paper studies attacking recommender systems on diverse open-ended Web platforms, relevant to user modeling and recommendation track.

## CCS CONCEPTS

• **Information systems** → **Recommender systems**.

## KEYWORDS

Recommender Attack, Target User Attack, Uplift Modeling, Treatment Effect Estimation

**ACM Reference Format:**
Anonymous Author(s). 2024. Uplift Modeling for Target User Attacks on Recommender Systems: Submission ID: 507. In *Proceedings of the ACM Web Conference 2024 (WWW '24), May 13-17 2024, Singapore.* ACM, New York, NY, USA, 17 pages.

## 1 INTRODUCTION

Recommender systems have evolved into fundamental services for information filtering on numerous Web platforms such as Amazon and Twitter. Recent research has validated the vulnerability

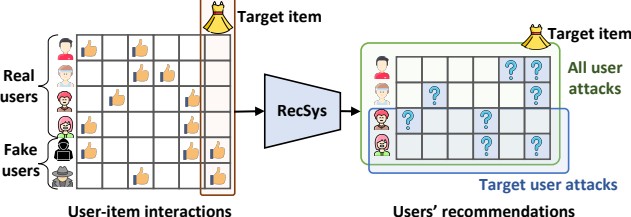

**(a) Illustration of all user attacks and target user attacks.**

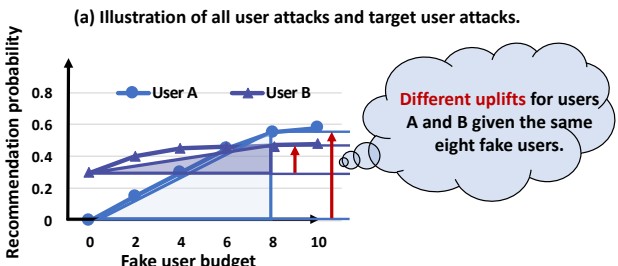

**(b) Recommendation probability of users *w.r.t.* varying fake user budgets.**

**Figure 1: Illustration of injective attacks on all users and target users (a) and varying attack difficulty on two users (b), where one fake user may cause different uplifts of the recommendation probabilities.**

of recommender systems to *injective attacks* [28, 67, 72], which aim to promote the exposure of a target item via injecting limited fake users (refer to Figure 1(a)). Specifically, since recommender models typically utilize Collaborative Filtering (CF) in users' historical interactions to make recommendations, the attackers can fabricate fake user interactions and inject them into open-world Web platforms, so as to induce recommender models to elevate the exposure probability of a target item. As a result, such injective attacks can deliberately amplify traffic to target items, potentially bringing economic, political, or other profits to certain entities.

Generally, past literature on injective attacks falls into three main groups: 1) *Heuristic attackers* [7, 24] that adopt heuristic rules to construct fake users, for instance, Bandwagon Attack [6] increases the co-occurrence of popular items and the target item in fake user interactions; 2) *gradient-based attackers* [16, 27] that directly adjust the interactions of fake users via gradients to maximize the well-designed attack objectives; and 3) *neural attackers* [31, 32, 44] that optimize the neural networks to generate influential fake users for promoting the target item to more users.

However, previous work neglects that not all users will be interested in the target item, as each item appeals to its unique audience. For instance, dresses typically appeal more to female buyers. Increasing the recommendations of a target item to all users not only wastes attack resources, but also results in inferior attack performance. Worse still, most studies fail to account for the varying attack difficulty across different users [28] (see Figure 1(b)). Some

"harder" users need more fake users to receive exposure to a target item. Due to ignoring the varying attack difficulty, easy users might receive redundant fake user budgets while hard users might be inadequately attacked, leading to inefficient use of budgets and poor attack performance.

To address the issues, we focus on an interesting recommender attack task — *target user attacks*, which attempt to expose a target item to a specific user group instead of all users. Moreover, we formulate the varying attack difficulty via causal language. From a causal view, assigning fake user budgets to a target user can be formulated as a treatment, and the probability of recommending a target item to this target user is the outcome. Given a target item, the varying attack difficulty essentially reflects the heterogeneous treatment effects (*a.k.a.* uplifts[1]) on different target users. This is attributed to the different similarities between the target item and each target user's historically liked items. In this light, the key to maximizing attack performance with limited fake user budgets lies in: 1) estimating the heterogeneous treatment effect on each target user with different budgets, and 2) allocating the limited budgets wisely to maximize the treatment effects across all target users, *i.e.*, the overall recommendation probability.

To this end, we present an Uplift-guided Budget Allocation (UBA) framework for target user attacks. In particular, UBA utilizes two methods to estimate the treatment effect. If a surrogate recommender model (*e.g.*, Matrix Factorization (MF)) is available to simulate the victim recommender model, UBA conducts simulation experiments with different budgets to attack target users, and then repeats the experiments to assess the treatment effect. In the absence of a reliable surrogate model, we identify a proxy variable for UBA to approximate the heterogeneous treatment effects between target users and items. We have empirically and theoretically validated that the identified proxy variable — the three-order path number between the target user and item in the user-item interaction graph — exhibits a strong positive correlation with the recommendation probability of CF models. Based on the estimated treatment effects, UBA employs a dynamic programming algorithm to compute the optimal budget allocation for all target users, maximizing the overall recommendation probability.

Since UBA is a model-agnostic framework, we instantiate it on three competitive attack models and conduct experiments on three benchmark datasets. Extensive experiments show the superiority and generalization ability of UBA in various settings, such as different target items, fake user budgets, attack models, and victim models. Moreover, we validate the robustness of UBA in the cases of applying defense models although the defense models are effective to some extent. To ensure reproducibility, we release our code and data at https://anonymous.4open.science/r/UBA-C52D.

To summarize, our contributions are threefold.

- We highlight the significance of target user attacks and formally inspect the issue of varying attack difficulty on users from a causal perspective.
- We propose the model-agnostic UBA framework, which offers two methods to estimate the heterogeneous treatment effects on target users and calculates the optimal budget allocation to maximize attack performance.

- Extensive experiments reveal the significance of UBA in performing target user attacks across diverse settings. Meanwhile, we validate the robustness of UBA against defense models.

## 2 RELATED WORK

In this section, we introduce closely related concepts and literature on uplift modeling and injective attacks. More related studies may refer to Appendix Section D.

• **Uplift Modeling.** Uplift, a term commonly used in marketing, usually represents the difference in purchasing actions between customers who receive a promotional offer (the treated group) and those who do not (the control group) [35, 68]. In causal language, uplift essentially quantifies the causal effect of a treatment (*e.g.*, a promotion) on the outcome (*e.g.*, purchasing behaviors). Despite extensive research on uplift modeling in the machine learning and marketing communities [1, 18, 46], the use of uplift modeling in recommendation receives little scrutiny [52, 54, 59]. Initial studies only consider the potential of uplift modeling to regulate the exposure proportion of item categories [62]. By contrast, we define the assigned fake user budgets in injective attacks as the treatment and estimate the difference of recommendation probabilities on target users as the uplifts. Based on the estimated uplifts, we aim to determine the best treatment for budget allocation to maximize overall attack performance.

• **Injective Attacks.** The objective of injective attacks (*a.k.a.* shilling attacks) is to promote the recommendations of a target item to all users on the recommender platform [7, 23, 47, 55, 65]. Given a target item, the attacker optimizes fake user interactions, and then the interactions of real users and generated fake users are fed into the victim model for training, improving the recommendation probabilities of the target item to all real users.

Formally, give a target item $i$ in the item set $\mathcal{I}$, and a set of real users $\mathcal{U}_r$ with their historical interaction matrix $D_r \in \{0,1\}^{|\mathcal{U}_r| \times |\mathcal{I}|}$ where 1 and 0 indicate users' liked and disliked items, the attacker aims to craft $D_f \in \{0,1\}^{|\mathcal{U}_f| \times |\mathcal{I}|}$, the interaction matrix of a set of fake users $\mathcal{U}_f$, for maximizing the attack objective $O$ on a victim recommender model $\mathcal{M}_\theta$:

$$\max_{D_f} O(\mathcal{M}_{\theta^*}, \mathcal{U}_r, i),$$
$$\text{s.t. } \theta^* = \arg\min_\theta \mathcal{L}(\mathcal{M}_\theta, D); |\mathcal{U}_f| \leq N, \tag{1}$$

where the victim recommender model $\mathcal{M}_{\theta^*}$ is well trained via the loss function $\mathcal{L}(\cdot)$ calculated on the interactions of both real and fake users, *i.e.*, $D = \begin{bmatrix} D_r \\ D_f \end{bmatrix}$. Besides, the budget of fake users is limited by a hyper-parameter $N$ [27, 64, 65]. In particular, we detail three key components, attack objective, attack knowledge, and optimization of $D_f$, as follows.

***Attack objective.*** The attack objective $O(\mathcal{M}_{\theta^*}, \mathcal{U}_r, i)$ is usually defined as enhancing the probabilities of recommending the target item $i$ to all users $\mathcal{U}_r$ by the victim model $\mathcal{M}_{\theta^*}$. Generally, it can be evaluated by the hit ratio on the real user set $\mathcal{U}_r$, where a user is "hit" only when the target item $i$ is ranked into this user's Top-$K$ recommendation list.

---

[1] Treatment effects and uplifts are exchangeable in this work [1, 68] .

***Attack knowledge.*** Existing studies have different assumptions for the knowledge available to the attacker, where the knowledge mainly involves the users' interaction data $D_r$ and the victim recommender model $\mathcal{M}_{\theta^*}$. Specifically, white-box settings [15, 16, 27] might presume both $D_r$ and the parameters of $\mathcal{M}_{\theta^*}$ are accessible to the attacker. By contrast, the definitions of gray-box settings vary [9, 50, 58]. While they consistently assume the accessible $D_r$, the usage of $\mathcal{M}_{\theta^*}$ differs [14, 43, 44, 48, 64, 69]. Some work utilizes the recommendation lists from $\mathcal{M}_{\theta^*}$ [14, 43] while some researchers assume $\mathcal{M}_{\theta^*}$ is totally unavailable and only adopt a surrogate model $\mathcal{S}_\phi$ as a replacement for attacking [44, 48, 64, 69].

***Optimization of $D_f$.*** To adjust $D_f$ for maximizing the attack objective, existing methods fall into three lines [12, 37, 42]. First, heuristic attackers intuitively increase the co-occurrence of some selected items and the target item $i$ in $D_f$ via some heuristic rules, enhancing the popularity of item $i$ [28]. However, such methods cannot directly optimize the attack objectives, leading to poor attack performance [16, 27, 29]. Besides, to maximize the attack objectives, gradient-based methods directly optimize the interactions of fake users [16, 27, 29] while neural attackers optimize neural networks to generate fake user interactions [23, 31, 32, 44]. Their optimization process typically utilizes the recommendations of the victim model or a surrogate model for gradient descent [39, 56].

## 3 TASK FORMULATION

In this section, we formulate the task of target user attacks. Besides, we quantify the attack difficulty across users from a causal view.

● **Target User Attacks.** Despite the great success of injective attacks, we focus on a novel recommender attack task — target user attacks, which aim to promote the recommendations of a target item to a group of specific users. This is more reasonable since each target item has its own potential audience. Blindly attacking all users will waste the limited fake user budgets and lead to suboptimal attack performance. Under target user attacks, the attacker can freely specify potential users for attacking based on user features, users' historical interactions, or attack difficulty.

***Attack objective.*** Formally, target user attacks change the objective $O(\mathcal{M}_{\theta^*}, \mathcal{U}_r, i)$ in Eq. (1) to $O(\mathcal{M}_{\theta^*}, \mathcal{U}_t, i)$, where $\mathcal{U}_t$ denotes the selected target user group.

***Attack knowledge.*** In this work, we adhere to a stricter yet more practical setting: only the interactions of a proportion of real users are accessible, given the fact that the attackers can never collect the interactions of all users. Furthermore, we assume the victim model $\mathcal{M}_{\theta^*}$ is unknown. We consider the situations with (w/) and without (w/o) using a surrogate model $\mathcal{S}_\phi$.

***Optimization of $D_f$.*** Previous methods of injective attacks can be adapted for target user attacks by revising the attack objective to $O(\mathcal{M}_{\theta^*}, \mathcal{U}_t, i)$. In detail, heuristic attackers [6, 7, 24, 33] are able to increase the co-occurrence of target items and the items liked by the target users. Gradient-based and neural attackers may optimize the generation of fake user interactions to maximize $O(\mathcal{M}_{\theta^*}, \mathcal{U}_t, i)$ [27, 31, 44]. For instance, neural attackers [32] can randomly sample a target user as a template for fake user generation to increase the similarity between fake users and the target user, and then forcibly set the fake user feedback on the target item as like, pushing the victim model to recommend the target item to the target user.

● **Attack Difficulty.** However, these intuitive methods ignore the varying attack difficulty on different target users. As illustrated in Figure 1(b), an increase in the fake user budgets from 4 to 5 might be negligible to uplift the recommendation probabilities on user B while it can significantly enhance that for user A. Gradient-based attackers directly optimize fake user interactions for attacking all target users while neural attackers may randomly sample target users as templates to generate fake users by neural networks. They are averagely assigning fake user budgets to all target users without considering the varying attack difficulty. As a special case, DADA [62] considers the difficulty of attacking each user regarding a target item, yet it only employs a greedy algorithm to emphasize easier users than difficult users. In this way, DADA fails to quantify the attack difficulty across users and achieve optimal budget allocation to target users.

Formally, we measure the attack difficulty of a target user $u$ regarding a target item $i$ via a function of recommendation probability *w.r.t.* the fake user interactions $D_f$, denoted as $Y_{u,i}^{\theta^*}(D_f)$, which represents the probability of ranking item $i$ into the Top-$K$ recommendation list of user $u$ by the victim model $\mathcal{M}_{\theta^*}$. Notably, the fake user interaction matrix $D_f$ in $Y_{u,i}^{\theta^*}(\cdot)$ is determined by the attacker with the fake user budget allocation $T$. From a causal view, we formulate the fake user budget allocation on target users as a multi-dimension treatment $T = \{t_u\}_{u \in \mathcal{U}_t}$, where $t_u$ satisfies that $\sum_{u \in \mathcal{U}_t} t_u \leq N, t_u \geq 0$, and $t_u \in \mathbb{Z}$. Given a target item $i$, a victim model $\mathcal{M}_{\theta^*}$, and fake user interactions $D_f$ with treatment $T$, the outcome across all target users is formulated as $\{Y_{u,i}^{\theta^*}(D_f(T))\}_{u \in \mathcal{U}_t}$.

Given a target item $i$, a set of target users $\mathcal{U}_t$, and $N$ fake user budgets, the objective of target user attacks can be reformulated to consider attack difficulty as follows:

$$\max_{T, D_f} O(\mathcal{M}_{\theta^*}, \mathcal{U}_t, i) = \sum_{u \in \mathcal{U}_t} Y_{u,i}^{\theta^*}(D_f(T)),$$
$$\text{s.t.} \sum_{u \in \mathcal{U}_t} t_u \leq N; t_u \geq 0, t_u \in \mathbb{Z} \text{ for any } u \in \mathcal{U}_t, \tag{2}$$

where we optimize the fake user interaction matrix $D_f$ to maximize the overall Top-$K$ recommendation probability on all target users. $D_f$ is further affected by different attackers[2] and the budget allocation $T$ across target users, which considers varying attack difficulty. For instance, given a target user $u$ with a budget $t_u$, heuristic attackers can construct $t_u$ fake users with similar interactions to user $u$, increasing the co-occurrence probability of target items and the items liked by user $u$. Gradient-based attackers can optimize the interactions of $t_u$ fake users specifically to maximize the attack objective of target user $u$. Neural attackers might take the target user $u$ as a template to generate the interactions of $t_u$ fake users. Beyond averagely assigning fake user budgets, the attackers can execute target user attacks more purposefully and effectively by considering varying attack difficulty for treatment $T$.

To keep the generality and simplicity of the optimization, we aim to design a model-agnostic solution to first estimate the optimal budget allocation $T^*$ for a given attacker, and then we can apply $T^*$ to the attacker for the final optimization of $D_f$.

---

[2]To keep notation brevity, we omit the notation of attackers in $D_f(T)$.

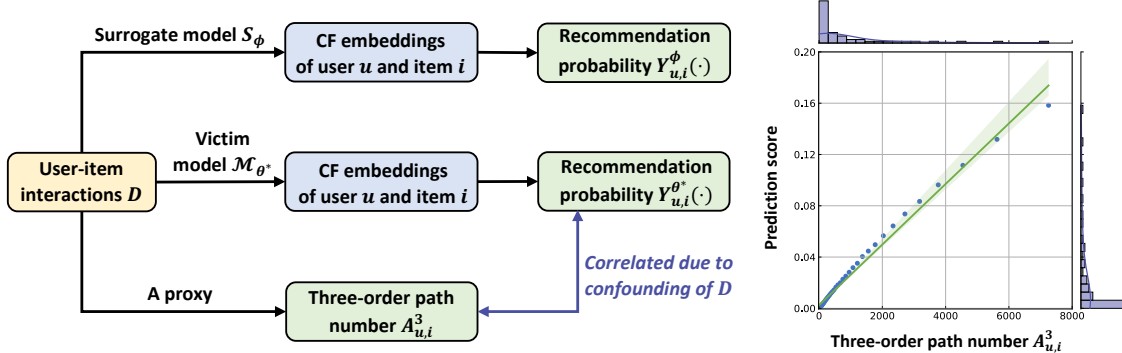

(a) Illustration of UBA to estimate $Y_{u,i}^{\theta^*}(\cdot)$ w/ and w/o a surrogate model.     (b) Correlation visualization when $\mathcal{M}_{\theta^*}$ is MF.

**Figure 2: Illustration of two estimation methods and the correlation analysis from a causal view, and (b) correlation visualization between three-order path numbers of user-item pairs and their prediction scores by MF, where the correlation coefficients $r = 0.9998 \approx 1$ and $p = 6e^{-86} \ll 0.001$ via the Spearman Rank Correlation Test [63] validate the strong correlation (see similar correlation on more CF models in Appendix Section A.1).**

• **Objective.** The key to considering the varying attack difficulty for the maximal attack objective lies in 1) estimating $Y_{u,i}^{\theta^*}(D_f(t_u))$ of each target user $u$ *w.r.t.* different budgets $t_u$[3] (see the curves in Figure 1(b)), and 2) calculating the optimal treatment $T^*$ to allocate fake users and generate $D_f$ for attacking.

## 4 UPLIFT-GUIDED BUDGET ALLOCATION

We propose a UBA framework with two methods to estimate the treatment effects $Y_{u,i}^{\theta^*}(D_f(t_u))$ on each target user, and then calculate the optimal treatment $T^*$ for target user attacks. Lastly, we detail how to instantiate UBA on existing attackers.

### 4.1 Treatment Effect Estimation

Since the victim model $\mathcal{M}_{\theta^*}$ in Eq. (2) is unknown to attackers, we propose two methods w/ and w/o the surrogate model $\mathcal{S}_\phi$ to estimate $Y_{u,i}^{\theta^*}(D_f(t_u))$ *w.r.t.* varying budgets $t_u$.

• **Estimation via Simulation Experiments.** If $\mathcal{S}_\phi$ is available to replace $\mathcal{M}_{\theta^*}$, UBA can utilize $\mathcal{S}_\phi$ to do simulation experiments for the estimation of $Y_{u,i}^{\theta^*}(D_f(t_u))$ with varying $t_u$. In detail, we repeat $E$ times of simulation experiments with different random seeds. In each experiment, given a target item $i$, an attacker, and a surrogate model $\mathcal{S}_\phi$, we do the following steps: 1) assigning the same $t_u$ fake users to all target users; 2) using the attacker to generate the fake user matrix $D_f$ based on the assigned budget; and 3) leveraging $D_f$ to attack $\mathcal{S}_\phi$ and obtain its Top-$K$ recommendations for each target user. After $E$ times of experiments with different random seeds for training ($E \approx 10$ in our implementation), we can approximate $Y_{u,i}^{\theta^*}(D_f(t_u))$ via the hit ratio of target user $u$:

$$Y_{u,i}^{\theta^*}(D_f(t_u)) \approx \frac{E'_{t_u}}{E}, \tag{3}$$

where $E'_{t_u}$ is the number of successfully promoting the target item $i$ into the Top-$K$ recommendations of target user $u$ by using $t_u$ budgets in $E$ times of experiments. Similarly, we can vary $t_u \in$

$\{1, 2, ..., H\}$ with $H \ll N$ to estimate $Y_{u,i}^{\theta^*}(D_f(t_u))$ with different fake user budgets[4].

• **Estimation via High-order Interaction Path.** Although UBA can estimate the treatment effects via simulation experiments, they rely on a reliable surrogate model and require repeated simulation experiments. To shake off these shackles, we propose another treatment effect estimation method w/o using surrogate models and simulation experiments.

We analyze the reasons for the varying attack difficulty across target users. Recommender models, including $\mathcal{M}_{\theta^*}$ and $\mathcal{S}_\phi$, learn CF embeddings of users and items from interactions as illustrated in Figure 2(a), and then measure the similarity between their CF embeddings to output prediction scores for item ranking. The varying attack difficulty is attributed to the distinct CF embeddings, and essentially stems from the different similarities *w.r.t.* interactions. To eliminate the dependence on surrogate models, we consider finding a proxy of interactions for the estimation of $Y_{u,i}^{\theta^*}(D_f(t_u))$. We inspect the number of high-order paths between a target user and a target item in the user-item interaction graph, discovering that it has a strong positive correlation with the user-item prediction scores as shown in Figure 2(b). Formally, if we define $A = \begin{bmatrix} 0 & D_r \\ D_r^T & 0 \end{bmatrix}$, the three-order path number is the value in $A^3$ indexed by a user and an item, which describes the easiness of connecting this user-item pair via some intermediate users and items (see more explanation in Appendix Section A.1).

PROPOSITION 1. *Given a user and an item without historical interactions, their prediction score by a CF model is positively correlated with their three-order[5] path number in $A^3$.*

We theoretically analyze the rationality and robustness of Proposition 1 *w.r.t.* different CF models in Appendix Section A.2. In Appendix Section A.1, we also present extensive experiments to validate the wide existence of such a correlation.

---

[3]Here we change $T$ in $Y_{u,i}^{\theta^*}(D_f(\cdot))$ to $t_u$ to denote how $Y_{u,i}^{\theta^*}(\cdot)$ of target user $u$ changes *w.r.t.* the user $u$' own budget $t_u$. The mutual interference of fake user budgets among target users is tough to measure, which is left for further exploration.

[4]The potential budget of each target user is far less than the total budget number $N$ because $N$ budgets are assigning to a group of target users.

[5]We find that the correlation exists with multiple different order numbers (see Appendix Section A.1). To keep simplicity, we select the smallest order for investigation.

Intuitively, CF models utilize the interaction similarity across users and items for recommendations, and three-order path numbers describe the closeness between users and items in the interaction graph. The user-item pairs with larger three-order path numbers are intuitively closer to each other in the CF embedding space, leading to higher prediction scores (see Figure 2(b)). Since the prediction scores are used to rank Top-$K$ recommendations, the three-order path number is also positively correlated with the Top-$K$ recommendation probability $Y_{u,i}^{\theta^*}(\cdot)$. Moreover, from a causal view in Figure 2(a), user-item interactions act as a confounder, causing the correlation between three-order path numbers and Top-$K$ recommendation probabilities.

Thanks to such a positive correlation, we can approximate $Y_{u,i}^{\theta^*}(\cdot)$ by the weighted three-order path number between user $u$ and item $i$, denoted as $\alpha \cdot (A_{u,i}^3)^\beta$, where $\alpha$ and $\beta$ are two hyper-parameters to adapt for the correlation. Thereafter, to estimate $Y_{u,i}^{\theta^*}(D_f(t_u))$ with varying budgets $t_u$, we only need to assess $\alpha \cdot (A_{u,i}^3)^\beta$ after injecting $t_u$ fake users to maximize the three-order path number between $u$ and $i$. By analyzing the relationships between fake user interactions and the three-order path number, we have the following finding.

PROPOSITION 2. *The three-order path number between target user $u$ and target item $i$ ($A_{u,i}^3$) is equivalent to the weighted sum of the intermediate users who liked item $i$ in $A$, where the weights are their interaction similarities with target user $u$, i.e., the number of mutually liked items.*

Please refer to Appendix Section A.3 for the proofs of Proposition 2. In the light of Proposition 2, given a target user $u$ and a target item $i$, we can construct $t_u$ fake users with the largest interaction similarities with target user $u$ and a like interaction with target item $i$. Consequently, we can obtain the optimal three-order path number in $(A')^3$ to estimate $Y_{u,i}^{\theta^*}(D_f(t_u))$ via $Y_{u,i}^{\theta^*}(D_f(t_u)) \approx \alpha \cdot ((A')_{u,i}^3)^\beta$, where $A' = \begin{bmatrix} 0 & 0 & D_r \\ 0 & 0 & D_f \\ D_r^T & D_f^T & 0 \end{bmatrix}$, *i.e.,* the symmetric interaction matrix with both real and fake users.

### 4.2 Budget-constrained Treatment Optimization

After estimating the treatment effect $Y_{u,i}^{\theta^*}(D_f(t_u))$, the selection of treatment $T$ in Eq. (2) becomes a budget-constrained optimization problem [1]. To calculate the optimal treatment $T^*$, we implement a dynamic programming algorithm based on the idea of the knapsack problem (see Appendix Section A.4 for details). Afterward, we can allocate optimal budgets to each target user to enhance existing attackers for superior overall attack performance.

● **Instantiation on Existing Attackers.** Given the optimal $T^*$, existing attackers, including heuristic, gradient-based, and neural attackers, can allocate fake users accordingly and utilize their own strategies to construct fake user interactions (see analysis of Eq. (2) in Section 3 for details). These fake users are subsequently used to attack victim recommender models.

● **UBA for Enhancing Recommendation Security.** A superior and transparent attacker can always inspire stronger defense methods. Recommender platforms can utilize UBA to improve their

**Table 1: Statistics of the three datasets.**

| | $|\mathcal{U}_r|$ | $|\mathcal{I}|$ | #Interactions | Sparsity | K-core |
|---|---|---|---|---|---|
| **ML-1M** | 5950 | 3702 | 567,533 | 0.257% | 10 |
| **Amazon** | 3179 | 5600 | 38, 596 | 0.216% | 10 |
| **Yelp** | 54632 | 34474 | 1,334,942 | 0.070% | 10 |

defense models through adversarial training, thereby enhancing the security of recommendations.

## 5 EXPERIMENTS

In this section, we conduct extensive experiments to answer the following research questions:

– **RQ1:** How does UBA enhance existing attackers in the task of target user attacks?

– **RQ2:** How does UBA generalize across various settings (*e.g.,* different victim models, budgets, and accessible interactions)?

– **RQ3:** How do UBA and other attackers perform if defense models are applied?

● **Datasets and Metric.** We evaluate the attackers on the three real-world datasets: MovieLens-1M (ML-1M) [19], Amazon-Game (Amazon) [40], and Yelp [4]. Table 1 displays the statistics of the datasets. In addition to the commonly used ML-1M and Amazon datasets in previous recommender attack work [28, 31], we also select Yelp for experiments due to its larger scale and sparsity. In this way, we can analyze target user attacks in more diverse scenarios. We treat target items as the positive items and use a widely used metric Hit Ratio@$K$ (HR@$K$) [28] to measure how many target users receive target item recommendation, where $K = 10$ or $20$, denoting the length of recommendation lists. Additionally, we introduce NDCG@$K$ and MRR@$K$ to consider ranking positions of target items in the recommendation lists.

● **Data Processing.** For data processing, we conduct a 10-core filtering on all three datasets to ensure data quality. Besides, all three datasets consist of explicit user feedback such as ratings, which might be required by some existing attackers. We follow the default requirements of the attackers to provide explicit or implicit feedback. For the victim recommender models and surrogate recommender models, we employ the common implicit feedback with binary values $\{0, 1\}$ in recommender training. Following previous studies [22], we map historical interactions greater than 3 to likes with label 1 and the remaining to dislikes with label 0.

● **Baselines.** We compare with the following baselines: heuristic attackers including Random [24], Segment [6], Bandwagon [6], and Average Attacks [26], the state-of-the-art gradient-based attacker DADA with its variants [28], and neural attackers including WGAN [3], AIA [44], AUSH [31], and Leg-UP [32]. To adapt to target user attacks, we propose a model-agnostic baseline named "Target" by changing the attack objective from $O(\mathcal{M}_{\theta^*}, \mathcal{U}_r, i)$ to $O(\mathcal{M}_{\theta^*}, \mathcal{U}_t, i)$ as discussed in Section 3. We implement "Target" and our UBA framework to three competitive backend attackers for comparison. We move the hyper-parameter tuning to Appendix Sections B.1.

● **Victim and Surrogate Models.** Following [28], we choose representative MF [25], NCF [20], and LightGCN [22] as victim models and utilize the simplest MF as the surrogate model. Table 2 shows

**Table 2: Evaluation on three real-world datasets when overall fake user budget is 100. We show the results of HR@10, NDCG@10, and MRR@10. The best results for each backend model are bold and the second-best ones are underlined. ∗ implies the improvements over the best baseline "Target" are statistically significant ($p$-value<0.05) under $t$-test.**

| | ML-1M | | | | | | Yelp | | | | | | Amazon | | | | | |
|---|---|---|---|---|---|---|---|---|---|---|---|---|---|---|---|---|---|---|
| | Popular item | | | Unpopular item | | | Popular item | | | Unpopular item | | | Popular item | | | Unpopular item | | |
| | HR | NDCG | MRR | HR | NDCG | MRR | HR | NDCG | MRR | HR | NDCG | MRR | HR | NDCG | MRR | HR | NDCG | MRR |
| Beofore Attack | 0.0000 | 0.0000 | 0.0000 | 0.0000 | 0.0000 | 0.0000 | 0.0000 | 0.0000 | 0.0000 | 0.0000 | 0.0000 | 0.0000 | 0.0200 | 0.0035 | 0.0020 | 0.0000 | 0.0000 | 0.0000 |
| Random Attack | 0.0000 | 0.0000 | 0.0000 | 0.0000 | 0.0000 | 0.0000 | 0.0200 | 0.0028 | 0.0022 | 0.0200 | 0.0028 | 0.0022 | 0.2000 | 0.0235 | 0.0199 | 0.2200 | 0.0262 | 0.0220 |
| Segment Attack | 0.0200 | 0.0040 | 0.0022 | 0.0200 | 0.0035 | 0.0035 | 0.0000 | 0.0000 | 0.0000 | 0.0400 | 0.0039 | 0.0040 | 0.1600 | 0.0195 | 0.0160 | 0.1400 | 0.0180 | 0.0155 |
| Bandwagon Attack | 0.0200 | 0.0035 | 0.0020 | 0.0000 | 0.0000 | 0.0000 | 0.0200 | 0.0029 | 0.0025 | 0.0200 | 0.0027 | 0.0020 | 0.2000 | 0.0245 | 0.0222 | 0.1600 | 0.0204 | 0.0177 |
| Average Attack | 0.0000 | 0.0000 | 0.0000 | 0.0000 | 0.0000 | 0.0000 | 0.0200 | 0.0031 | 0.0029 | 0.0200 | 0.0029 | 0.0025 | 0.1800 | 0.0217 | 0.0200 | 0.1400 | 0.0180 | 0.0355 |
| WGAN | 0.0200 | 0.0039 | 0.0025 | 0.0000 | 0.0000 | 0.0000 | 0.0200 | 0.0029 | 0.0025 | 0.0200 | 0.0027 | 0.0020 | 0.1400 | 0.0180 | 0.0155 | 0.1800 | 0.0227 | 0.0225 |
| DADA-DICT | 0.0400 | 0.0071 | 0.0073 | 0.0800 | 0.0104 | 0.0089 | 0.0200 | 0.0036 | 0.0040 | 0.1200 | 0.0169 | 0.0150 | 0.4200 | 0.0517 | 0.0420 | 0.3200 | 0.0448 | 0.0400 |
| DADA-DIV | 0.0000 | 0.0000 | 0.0000 | 0.0400 | 0.0057 | 0.0044 | 0.0000 | 0.0000 | 0.0000 | 0.1400 | 0.0220 | 0.0199 | 0.2600 | 0.0333 | 0.0260 | 0.3200 | 0.0473 | 0.0457 |
| DADA | 0.0600 | 0.0112 | 0.0123 | 0.1200 | 0.0156 | 0.0171 | 0.0400 | 0.0053 | 0.0080 | 0.1600 | 0.0268 | 0.0266 | 0.5400 | 0.0667 | 0.0540 | 0.4400 | 0.0576 | 0.0488 |
| AIA | 0.0200 | 0.0061 | 0.0033 | 0.0200 | 0.0127 | 0.0111 | 0.0000 | 0.0000 | 0.0000 | 0.0600 | 0.0054 | 0.0032 | 0.2800 | 0.0375 | 0.0311 | 0.2800 | 0.0393 | 0.0350 |
| +Target | 0.2000 | 0.0244 | 0.0222 | 0.3800 | 0.0428 | 0.0356 | 0.0000 | 0.0000 | 0.0000 | 0.0600 | 0.0082 | 0.0075 | 0.5800 | 0.0780 | 0.0725 | 0.6400 | 0.0846 | 0.0711 |
| + UBA(w/o $\mathcal{S}_\phi$) | 0.2400 | 0.0315 | 0.0300 | 0.4400 | 0.0449 | 0.0400 | 0.0200 | 0.0029 | 0.0025 | 0.1200 | 0.0191 | 0.0269 | 0.6600 | 0.0960 | 0.0942 | 0.7400 | 0.1007 | 0.0925 |
| + UBA(w/ $\mathcal{S}_\phi$) | 0.2600* | 0.0431* | 0.0520* | 0.5800* | 0.0556* | 0.0514* | 0.1000* | 0.0178* | 0.0250* | 0.1400* | 0.0255* | 0.0279* | 0.7200* | 0.1115* | 0.1200* | 0.8200* | 0.1162* | 0.1171* |
| AUSH | 0.0200 | 0.0047 | 0.0050 | 0.0000 | 0.0000 | 0.0000 | 0.0000 | 0.0000 | 0.0000 | 0.0000 | 0.0000 | 0.0000 | 0.2600 | 0.0347 | 0.0288 | 0.3200 | 0.0448 | 0.0400 |
| +Target | 0.2400 | 0.0349 | 0.0340 | 0.3600 | 0.0531 | 0.0422 | 0.0800 | 0.0111 | 0.0100 | 0.1000 | 0.0119 | 0.0100 | 0.6400 | 0.0887 | 0.0800 | 0.7400 | 0.1064 | 0.1057 |
| + UBA(w/o $\mathcal{S}_\phi$) | 0.2800 | 0.0380 | 0.0467 | 0.4200 | 0.0646 | 0.0550 | 0.1000 | 0.0138 | 0.0142 | 0.0800 | 0.0137 | 0.0160 | 0.7400 | 0.1064 | 0.1057 | 0.7400 | 0.1137 | 0.1233 |
| + UBA(w/ $\mathcal{S}_\phi$) | 0.3200* | 0.0474* | 0.0533* | 0.5200* | 0.0902* | 0.0828* | 0.1400* | 0.0255* | 0.0279* | 0.1600* | 0.0376* | 0.0533* | 0.8600* | 0.1299* | 0.1433* | 0.8000* | 0.1142* | 0.1142* |
| Legup | 0.0400 | 0.0062 | 0.0054 | 0.1000 | 0.0024 | 0.0021 | 0.0600 | 0.0078 | 0.0067 | 0.0400 | 0.0043 | 0.0050 | 0.0400 | 0.0063 | 0.0050 | 0.2000 | 0.0235 | 0.0199 |
| +Target | 0.1400 | 0.0187 | 0.0200 | 0.3200 | 0.0539 | 0.0412 | 0.1600 | 0.0251 | 0.0229 | 0.1800 | 0.0302 | 0.0300 | 0.6000 | 0.0742 | 0.0600 | 0.5000 | 0.0676 | 0.0625 |
| + UBA(w/o $\mathcal{S}_\phi$) | 0.1800 | 0.0239 | 0.0225 | 0.3200 | 0.0604 | 0.0525 | 0.2000 | 0.0283 | 0.0222 | 0.2600 | 0.0411 | 0.0433 | 0.7400 | 0.0961 | 0.0822 | 0.6800 | 0.0887 | 0.0755 |
| + UBA(w/ $\mathcal{S}_\phi$) | 0.2600* | 0.0480* | 0.0371* | 0.3600* | 0.0825* | 0.0743* | 0.2400* | 0.0365* | 0.0342* | 0.2800* | 0.0411* | 0.0466* | 0.8600* | 0.1216* | 0.1228* | 0.7600* | 0.1086* | 0.1085* |

the results of LightGCN as the victim model due to space limitation. The robustness of attacking MF and NCF is analyzed in Figure 4.

• **Selection of Target Items.** To verify the robustness of UBA on different target items, we test UBA on popular items and unpopular items, respectively. This is because the attack performance on popular and unpopular items might vary. We divide all items into five groups according to their popularity, *i.e.*, historical interaction numbers, and then we randomly select the popular item and the unpopular item as the target item from the most popular group and the most unpopular group, respectively. For each target item, we run five attack processes by changing random seeds and report the average performance.

• **Selection of Target Users.** Notably, target users can be specified by user IDs, attributes (*e.g.*, gender), and interactions. In this work, we try to find out the potential target users who might be interested in the target item. To achieve this, we select all users who have interacted with the target item category. For example, given an action movie as the target item, we find out all the users who have interacted with at least one action movie. Thereafter, we rank all the selected users via their interaction number with the target item category. Only the users with an interaction number with the target item category smaller than a threshold (10) are further selected, from which we randomly sample some users who are hard to attack into the target user group. In this way, we can select some target users who are possibly interested in this item category yet hard to attack. We usually select 50 target users in our experiments while we analyze the effect of different target user numbers in Table 4.

## 5.1 Target User Attack Performance

• **Overall Comparison (RQ1).** In Table 2, we report the attack performance on target users. As an extension, we present the attack results of all users in Appendix Section C.1. From Table 2, we have the following findings.

1) **Superiority of DADA and "Target".** Most attackers (heuristic and neural ones) cannot achieve satisfactory performance on target user attacks. By contrast, DADA with variants and "Target" exhibit superior results, revealing the importance of distinguishing users for attacking. DADA and DADA-DICT utilize a greedy algorithm to allocate more budgets to easy users while "Target" concentrates fake user budgets to attack target users.

2) **Effectiveness of UBA.** Both UBA w/ and w/o $\mathcal{S}_\phi$ significantly enhance the attack performance of three backend attackers (AIA, AUSH, and Leg-UP) by a large margin on three datasets. In addition, UBA also surpasses "Target", further validating the superiority of considering the varying attack difficulty. Due to estimating the attack difficulty across target users, UBA can rationally allocate the fake user budgets to maximize the overall recommendation probabilities. The users with large uplifts of recommendation probabilities will be favored, thus enhancing the attack performance.

3) **UBA(w/ $\mathcal{S}_\phi$) outperforms UBA(w/o $\mathcal{S}_\phi$).** This is reasonable, as the surrogate model $\mathcal{S}_\phi$ can assist in accurately estimating the attack difficulty through simulation experiments. $\mathcal{S}_\phi$ may serve as a reliable substitute for the victim model $\mathcal{M}_{\theta^*}$ in such estimation because they commonly leverage CF information for recommendations. Despite the better performance of UBA(w/ $\mathcal{S}_\phi$), UBA(w/o $\mathcal{S}_\phi$) does not need the simulation experiments, reducing the computation costs. As such, UBA(w/o $\mathcal{S}_\phi$) is also a favorable approach with effectiveness and efficiency.

• **Varying Fake User Budgets (RQ2).** Figure 3 depicts the attack performance on ML-1M with varying budgets. The results on Amazon and Yelp with similar trends are omitted to save space. From the figure, we can find that: 1) under different budgets, UBA w/ and w/o $\mathcal{S}_\phi$ usually achieve better attack performance than the original method and "Target" on three backend attackers. This verifies the robustness of UBA *w.r.t.* fake user budgets. And 2) the

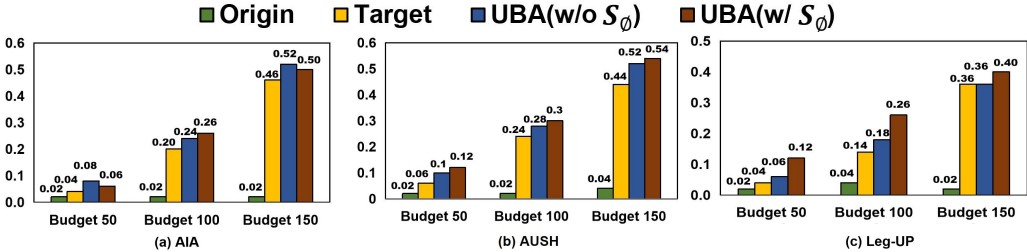

Figure 3: Performance comparison *w.r.t.* HR@10 under different attack budgets.

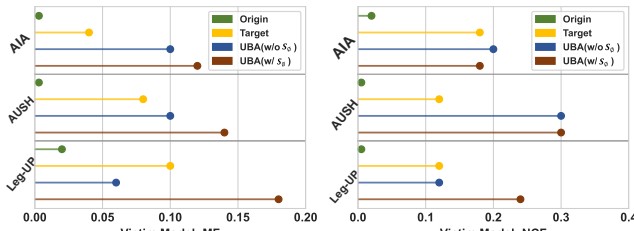

Figure 4: Generalization of UBA *w.r.t.* HR@10 across different victim models.

relative improvements from "Target" to UBA is significantly large when the budget is small such as budget= 50. Such observation on three attackers demonstrates that UBA holds high practical value in real-world recommender attack scenarios, given that the number of fake users an attacker can manage is often very limited [28].

• **Different Victim Models (RQ2).** Figure 4 visualizes the attack results of using MF and NCF as victim models on ML-1M. By inspecting Figure 4 and the results of LightGCN in Table 2, we can have the following observations. 1) UBA shows better performance than "Origin" and "Target" in most cases, indicating good generalization ability of UBA across different victim models. Indeed, the generalization of UBA lies in the accurate estimation of the attack difficulty on target users, *i.e.,* the heterogeneous treatment effects. By exploiting the CF information with and without using surrogate models, our proposed UBA is theoretically applicable to most CF victim models. And 2) the hit ratios on MF are significantly smaller than those on NCF and LigntGCN. This phenomenon commonly exists across different backend attackers and attack strategies. We attribute the possible reason to that NCF and LightGCN utilize more advanced neural networks to fully exploit CF information in user-item interactions, resulting in a more severe fitting of fake user interactions. Consequently, the fake users lead to a higher attack performance on NCF and LightGCN. This finding is critical, inspiring us to consider the security issues when devising advanced recommender models.

• **Effect of the Proportions of Accessible Interactions (RQ2).** The proportion of accessible interactions refers to the ratio of user interactions that the attacker can access. Both the estimation of treatment effects and the generation of fake users by the attackers are using these real user interactions. In real-world attack scenarios, attackers can only collect a limited portion of real users' interactions. Therefore, using the interactions of partial users can demonstrate the practicality of the attackers. We demonstrate the effects of different proportions of interactions in Table 3. From this table, we

Table 3: Performance with different proportions of user interactions accessible to attackers.

| Data Ratio | 10% | | 20% | | 50% | | 80% | |
|---|---|---|---|---|---|---|---|---|
| | HR@10 | HR@20 | HR@10 | HR@20 | HR@10 | HR@20 | HR@10 | HR@20 |
| **Before Attack** | 0.02 | 0.02 | 0.00 | 0.02 | 0.00 | 0.02 | 0.00 | 0.02 |
| **Random Attack** | 0.00 | 0.02 | 0.00 | 0.00 | 0.00 | 0.02 | 0.02 | 0.02 |
| **Segment Attack** | 0.00 | 0.02 | 0.02 | 0.02 | 0.02 | 0.02 | 0.02 | 0.02 |
| **Bandwagon Attack** | 0.02 | 0.02 | 0.02 | 0.02 | 0.02 | 0.02 | 0.00 | 0.04 |
| **Average Attack** | 0.00 | 0.00 | 0.00 | 0.00 | 0.00 | 0.02 | 0.00 | 0.00 |
| **WGAN** | 0.00 | 0.02 | 0.02 | 0.02 | 0.00 | 0.02 | 0.00 | 0.02 |
| **DADA-DICT** | 0.02 | 0.04 | 0.04 | 0.04 | 0.00 | 0.04 | 0.04 | 0.04 |
| **DADA-DIV** | 0.02 | 0.02 | 0.00 | 0.04 | 0.02 | 0.04 | 0.02 | 0.08 |
| **DADA** | 0.04 | 0.04 | 0.06 | 0.10 | 0.06 | 0.12 | 0.08 | 0.12 |
| **AIA** | 0.00 | 0.02 | 0.02 | 0.02 | 0.02 | 0.02 | 0.02 | 0.06 |
| +Target | 0.12 | 0.28 | 0.20 | 0.38 | 0.22 | 0.36 | 0.20 | 0.42 |
| + UBA(w/o $\mathcal{S}_\phi$) | 0.22 | 0.36 | 0.24 | 0.40 | 0.22 | 0.42 | 0.28 | 0.44 |
| + UBA(w/ $\mathcal{S}_\phi$) | **0.22** | **0.42** | **0.26** | **0.44** | **0.30** | **0.48** | **0.34** | **0.48** |
| **AUSH** | 0.00 | 0.04 | 0.02 | 0.02 | 0.04 | 0.06 | 0.04 | 0.08 |
| +Target | 0.16 | 0.34 | 0.24 | 0.36 | 0.24 | 0.36 | 0.30 | 0.40 |
| + UBA(w/o $\mathcal{S}_\phi$) | 0.20 | 0.42 | 0.28 | 0.40 | 0.26 | 0.42 | 0.32 | 0.44 |
| + UBA(w/ $\mathcal{S}_\phi$) | **0.28** | **0.44** | **0.32** | **0.42** | **0.30** | **0.46** | 0.28 | **0.44** |
| **Leg-UP** | 0.02 | 0.02 | 0.04 | 0.06 | 0.04 | 0.06 | 0.06 | 0.10 |
| +Target | 0.10 | 0.14 | 0.14 | 0.26 | 0.12 | 0.30 | 0.14 | 0.32 |
| + UBA(w/o $\mathcal{S}_\phi$) | 0.16 | 0.22 | 0.18 | 0.32 | 0.16 | 0.36 | 0.20 | 0.36 |
| + UBA(w/ $\mathcal{S}_\phi$) | **0.20** | **0.28** | **0.26** | **0.34** | **0.30** | **0.36** | **0.32** | **0.44** |

can observe the followings: 1) as the proportion increases, the attack performance shows a growing trend, indicating that receiving more user interactions is beneficial for attackers. And 2) two UBA methods usually achieve better performance than the baselines, validating the robustness of UBA under varying proportions of accessible interactions.

• **Varying Numbers of Target Users (RQ2).** In Table 4, we evaluate the attack performance of increasing the number of target users while maintaining the same fake user budgets. From this table, we can observe that: 1) with increasing target user numbers, two UBA methods accomplish better attack results than the baselines, demonstrating that UBA can handle the attacks on different numbers of target users. 2) Furthermore, two UBA methods remain relatively high hit ratios with limited fake user budgets when the number of target users is large, thereby guaranteeing attack effectiveness even if attackers seek to enlarge the attack scope.

• **Case Study.** To intuitively understand how UBA effectively utilizes fake user budgets to enhance attack effectiveness, we conduct a case study. We select five target users from ML-1M to compare the difference in fake user allocation between "Target" and UBA. Specifically, in Figure 5, given a target item, we show the varying recommendation probabilities of five target users under different fake user budgets, estimated by UBA(w/ $\mathcal{S}_\phi$). It is worth noting that "Target" randomly allocates fake user budgets while UBA maximizes overall recommendation probabilities for budget allocation.

**Table 4: Performance with different numbers of target users.**

|  | Target user 50 | | Target user 100 | | Target user 500 | |
|---|---|---|---|---|---|---|
|  | HR@10 | HR@20 | HR@10 | HR@20 | HR@10 | HR@20 |
| **Beofore Attack** | 0.00 | 0.02 | 0.04 | 0.06 | 0.00 | 0.02 |
| **Random Attack** | 0.00 | 0.00 | 0.10 | 0.14 | 0.10 | 0.15 |
| **Segment Attack** | 0.02 | 0.02 | 0.02 | 0.12 | 0.15 | 0.17 |
| **Bandwagon Attack** | 0.02 | 0.02 | 0.14 | 0.20 | 0.13 | 0.18 |
| **Average Attack** | 0.00 | 0.00 | 0.10 | 0.13 | 0.10 | 0.11 |
| **WGAN** | 0.02 | 0.02 | 0.05 | 0.12 | 0.12 | 0.18 |
| **DADA-DICT** | 0.04 | 0.04 | 0.31 | 0.42 | 0.30 | 0.53 |
| **DADA-DIV** | 0.00 | 0.04 | 0.20 | 0.33 | 0.29 | 0.41 |
| **DADA** | 0.06 | 0.10 | 0.30 | 0.44 | 0.28 | 0.55 |
| **AIA** | 0.02 | 0.02 | 0.18 | 0.23 | 0.17 | 0.30 |
| **+Target** | 0.20 | 0.38 | 0.20 | 0.38 | 0.25 | 0.39 |
| **+ UBA(w/o $\mathcal{S}_\phi$)** | 0.24 | 0.40 | 0.33 | 0.51 | 0.31 | 0.47 |
| **+ UBA(w/ $\mathcal{S}_\phi$)** | **0.26** | **0.44** | **0.44** | **0.70** | **0.38** | **0.55** |
| **AUSH** | 0.02 | 0.02 | 0.23 | 0.32 | 0.20 | 0.31 |
| **+Target** | 0.24 | 0.36 | 0.31 | 0.45 | 0.28 | 0.36 |
| **+ UBA(w/o $\mathcal{S}_\phi$)** | 0.28 | 0.40 | 0.44 | 0.56 | 0.34 | 0.39 |
| **+ UBA(w/ $\mathcal{S}_\phi$)** | **0.32** | **0.42** | **0.59** | **0.73** | **0.39** | **0.55** |
| **Legup** | 0.04 | 0.06 | 0.22 | 0.36 | 0.21 | 0.31 |
| **+Target** | 0.14 | 0.26 | 0.30 | 0.33 | 0.26 | 0.36 |
| **+ UBA(w/o $\mathcal{S}_\phi$)** | 0.18 | 0.32 | 0.41 | 0.50 | 0.40 | **0.60** |
| **+ UBA(w/ $\mathcal{S}_\phi$)** | **0.26** | **0.34** | **0.55** | **0.62** | **0.43** | 0.59 |

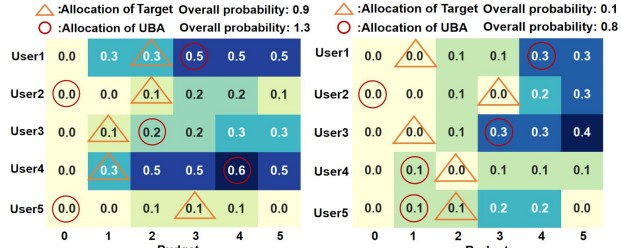

**Figure 5: Case study about the budget allocation on five target users. UBA allocates fake users more wisely to maximize the overall recommendation probability than Target.**

From Figure 5, we observe the following: 1) UBA allocates limited budgets to users with larger uplifts, leading to higher overall recommendation probabilities. For instance, for "User 1" in the left plot, increasing the budget from 2 to 3 results in an increase of 0.2 in the recommendation probability, significantly enhancing the probability of a successful attack. And 2) "Target", due to its random allocation of fake user budgets, usually assigns 1 to 3 fake user budgets to most users, leading to a wastage of fake user budgets on the users with constantly lower recommendation probabilities (*e.g.*, "User 2" and "User 4" in the right plot of Figure 5).

## 5.2 Defense Against Target User Attacks (RQ3).

In this section, we explore the ability of existing defense models against target user attackers. We examine two representative unsupervised defense models, PCA [36] and FAP [70]. PCA is the most classic model to detect a group of fake users for injective attacks and FAP is a unified framework for fake user detection based on a fraudulent action propagation algorithm. These defense models usually detect fake users and exclude the detected users for recommender training. Under the defense models, we present the attack results of three backend attackers with "Target" and UBA on ML-1M in Table 5.

**Table 5: Attack performance under two defense models.**

|  |  | AIA | | + Target | | + UBA(w/o $\mathcal{S}_\phi$) | | + UBA(w/ $\mathcal{S}_\phi$) | |
|---|---|---|---|---|---|---|---|---|---|
|  |  | Origin | Detector | Origin | Detector | Origin | Detector | Origin | Detector |
| **PCA** | **HR@10** | 0.02 | 0.02 | 0.20 | 0.14 | 0.26 | 0.20 | 0.24 | 0.22 |
|  | **HR@20** | 0.02 | 0.02 | 0.38 | 0.28 | 0.44 | 0.32 | 0.40 | 0.32 |
| **FAP** | **HR@10** | 0.02 | 0.02 | 0.20 | 0.08 | 0.26 | 0.10 | 0.24 | 0.12 |
|  | **HR@20** | 0.02 | 0.02 | 0.38 | 0.18 | 0.44 | 0.12 | 0.40 | 0.20 |
|  |  | **AUSH** | | + Target | | + UBA(w/o $\mathcal{S}_\phi$) | | + UBA(w/ $\mathcal{S}_\phi$) | |
| **PCA** | **HR@10** | 0.02 | 0.02 | 0.24 | 0.20 | 0.30 | 0.18 | 0.28 | 0.22 |
|  | **HR@20** | 0.02 | 0.02 | 0.36 | 0.36 | 0.42 | 0.32 | 0.40 | 0.34 |
| **FAP** | **HR@10** | 0.02 | 0.02 | 0.24 | 0.02 | 0.30 | 0.04 | 0.28 | 0.08 |
|  | **HR@20** | 0.02 | 0.02 | 0.36 | 0.10 | 0.42 | 0.10 | 0.40 | 0.14 |
|  |  | **Leg-UP** | | + Target | | + UBA(w/o $\mathcal{S}_\phi$) | | + UBA(w/ $\mathcal{S}_\phi$) | |
| **PCA** | **HR@10** | 0.04 | 0.00 | 0.14 | 0.12 | 0.26 | 0.12 | 0.18 | 0.14 |
|  | **HR@20** | 0.06 | 0.00 | 0.26 | 0.24 | 0.34 | 0.26 | 0.32 | 0.24 |
| **FAP** | **HR@10** | 0.04 | 0.00 | 0.14 | 0.04 | 0.26 | 0.06 | 0.18 | 0.08 |
|  | **HR@20** | 0.06 | 0.00 | 0.26 | 0.08 | 0.34 | 0.08 | 0.32 | 0.14 |

From the table, we can observe that: 1) both PCA and FAP decrease the performance of all attackers, indicating their usefulness in defending target user attackers. In particular, FAP achieves superior defense than PCA, attributable to its proficient propagation algorithm on the user-item interaction graph. However, 2) even with the defense models, UBA w/ and w/o $\mathcal{S}_\phi$ show generally higher hit ratios than the vanilla attackers and "Target". This possibly validates the capacity of UBA to allocate budgets to target users who are more interested in the target item, making the detection of fake users challenging. Future research could benefit from these two observations, especially inspirations from FAP, to devise defense models specially tailored for target user attackers.

• **More Experimental Results.** Due to space limitations, we provide more experimental analysis in the Appendix. First, in addition to target user attacks, we present the attack results on all users in Section C.1. We study the results of different victim models and defense models on all users in Section C.2 and Section C.3. Besides, we only report the performance of promoting popular items in Figure 3 to save space. The results on unpopular items are in Section C.4.

## 6 CONCLUSION AND FUTURE WORK

In this work, we highlighted the significance of target user attacks and formulated the issue of varying attack difficulty across users via causal language. To consider the varying attack difficulty and maximize attack performance, we proposed a model-agnostic UBA framework to calculate the optimal allocation of fake user budgets. We conducted extensive experiments on three real-world datasets with diverse settings, revealing the effectiveness of UBA in performing target user attacks. Moreover, we validated the robustness of UBA against defense models. This work also emphasizes the imperative for further exploration of target user attacks and the corresponding defense strategies.

As an initial study on uplift-enhanced recommender attack, this work leaves many promising future directions. First, we implement UBA on three attackers while UBA can be instantiated on more injective attackers to enhance their attack performance. Second, while we have confirmed the effectiveness of two defense models to some extent, it is highly valuable to design more advanced defense strategies specifically tailored for target user attackers. Lastly, we assume that the victim recommender models leverage CF information. Despite the popularity of CF models, UBA can be extended to attack non-CF models such as popularity-based models.

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

## Appendix

## A  METHOD

We validate the correlation of Proposition 1 via experiments in Section A.1, and present the theoretical analysis of Proposition 1 and Proposition 2 in Section A.2 and Section A.3, respectively. Lastly, we detail the dynamic programming algorithm for treatment selection in Section A.4.

### A.1  Correlation Analysis via Experiments

We conduct many experiments to verify the robustness of Proposition 1 on multiple CF models with two representative loss functions, BCE and BPR. In detail, we collect all user-item pairs without interactions in the training data and rank them via their high-order path numbers (see illustration in Figure 6), which are calculated via matrix multiplication over $A$. Besides, we obtain the prediction scores of these user-item pairs by different CF models. The ranked user-item pairs are split into 50 groups and we report the average prediction score and high-order path number of each group by scatter plots as shown in Figure 7 and Figure 8.

In particular, Figure 7 shows the robustness of the correlation between the three-order path number and the prediction score on three CF models (*i.e.,* MF, NCF, and LightGCN) with two popular recommender loss functions (*i.e.,* BCE and BPR). Furthermore, we validate that such correlation also holds on different orders such as five-order and seven-order path numbers as depicted in Figure 8. Besides, we conduct the Spearman Rank Correlation Test [63] for each figure. The correlation coefficients are all larger than 0.998 and the $p$-values are smaller than $1.46e^{-59} \ll 0.001$, indicating the strong positive correlations in these experiments. Last but not least, in Figure 7, we normalize the prediction scores via sigmoid for better visualization. Accordingly, we re-scale the three-order path numbers in some figures by $(A_{u,i}^3)^{0.3}$, where similar trends can also be obtained by the log function. From Figure 7, we can find that the correlation between the sigmoid-normalized prediction scores and the scaled three-order path numbers is approximately linear on LightGCN and the models with BPR loss. This shows the generality and robustness of using $\alpha \cdot (A_{u,i}^3)^\beta$ to approximate the treatment effect $Y_{u,i}^{\theta^*}(\cdot)$ in Section 4.1.

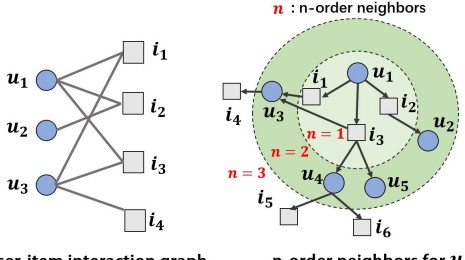

**Figure 6: Illustration of three-order neighbors on the user-item interaction graph.**

### A.2  Theoretical Analysis of Proposition 1

**Reminder of Proposition 1.** Given a user and an item without historical interaction, their prediction score by a CF model is positively correlated with their three-order path number in $A^3$.

PROOF. Formally, we reformulate our proposition as follows:

For different victim recommender models $\mathcal{M}_\theta$ with different loss functions $\mathcal{L}$, after sufficient training of the model parameters $\theta$, we have the positive correlation between the prediction score $\mathcal{M}_{\theta^*}(u, i)$ of a specific user-item pair $(u, i)$ and their three-order path number $A_{u,i}^3$:

$$\mathcal{M}_{\theta^*}(u, i) \propto A_{u,i}^3, \tag{4}$$

where $\theta^* = \arg\min_\theta \mathcal{L}(\mathcal{M}_\theta, D)$.

To begin with, we will prove our proposition via a specific example with MF and BPR loss function, and then demonstrate how our conclusion can generalize to other CF models and loss. To prove the proposition via the specific example, we first delve into the optimization process of the model, which can be formulated as:

$$\begin{aligned}
&\min_\theta \mathcal{L}_{\text{BPR}}(\mathcal{M}_\theta, D) \\
&= \frac{1}{|P|} \sum_{(u,i) \in P} \frac{1}{|\mathcal{I}|} \sum_{j \in \mathcal{I}} -\log(\sigma(\mathcal{M}_\theta(u, i) - \mathcal{M}_\theta(u, j))),
\end{aligned} \tag{5}$$

where $P$ denotes the positive interaction (positive user-item pair) set of $D$. $|P|$ represents the size of $P$ and $\sigma(t) = 1/(1 + \exp(-t))$. When we utilize MF as the recommender model, we have the prediction score $\mathcal{M}_\theta(u, i) = (e_u^\theta)^T (e_i^\theta)$, where the $e_u^\theta$ and $e_u^\theta$ denotes the learned embeddings of user $u$ and item $i$, respectively.

Taking the partial derivative of $\mathcal{L}_{\text{BPR}}$ with respect to the embedding of user $u^*$, we get:

$$\begin{aligned}
\frac{\partial \mathcal{L}_{\text{BPR}}(\mathcal{M}_\theta, D)}{\partial e_{u^*}^\theta} &= \frac{1}{|P|} \sum_{i \in \mathcal{N}_{u^*}} \frac{1}{|\mathcal{I}|} \sum_{j \in \mathcal{I}} \phi_\theta(u^*, i, j) \cdot (e_i^\theta - e_j^\theta) \\
&\approx \frac{1}{|P|} \sum_{i \in \mathcal{N}_{u^*}} \phi_\theta(u^*, i, j) \cdot e_i^\theta \\
&\quad - \frac{1}{|P|} \frac{|\mathcal{N}_{u^*}|}{|\mathcal{I}|} \sum_{j \in \mathcal{I}} \phi_\theta(u^*, i, j) \cdot e_j^\theta,
\end{aligned} \tag{6}$$

where $\mathcal{N}_{u^*}$ denotes the first-order neiborhood set of user $u^*$ and $|\mathcal{N}_{u^*}|$ represents its size. We utilize function $\phi_\theta(u, i, j)$ to represent $-[1 - \sigma(\mathcal{M}_\theta(u, i) - \mathcal{M}_\theta(u, j))]$, which is assumed approximately as the same value for different $(u, i, j)$ pairs. And $\phi_\theta(u, i, j)$ becomes close to zero as the training progresses. The reason for this approximation is that we assume that the MF model can well memorize or fit the training data.

Similarly, we obtain the partial derivative of the loss function $\mathcal{L}_{\text{BPR}}$ with respect to the embedding of item $i^*$ as follows:

$$\begin{aligned}
\frac{\partial \mathcal{L}_{\text{BPR}}(\mathcal{M}_\theta, D)}{\partial e_{i^*}^\theta} &= \frac{1}{|P|} \sum_{u \in \mathcal{N}_{i^*}} \frac{1}{|\mathcal{I}|} \sum_{j \in \mathcal{I}} \phi_\theta(u, i^*, j) \cdot e_u^\theta \\
&\quad - \frac{1}{|P|} \frac{1}{|\mathcal{I}|} \sum_{(u',i) \in P} \phi_\theta(u', i, i^*) \cdot e_{u'}^\theta \\
&\approx \frac{1}{|P|} \sum_{u \in \mathcal{N}_{i^*}} \phi_\theta(u, i^*, j) \cdot e_u^\theta \\
&\quad - \frac{1}{|P|} \frac{1}{|\mathcal{I}|} \sum_{u' \in \mathcal{U}} |\mathcal{N}_{u'}| \cdot \phi_\theta(u', i, i^*) \cdot e_{u'}^\theta,
\end{aligned} \tag{7}$$

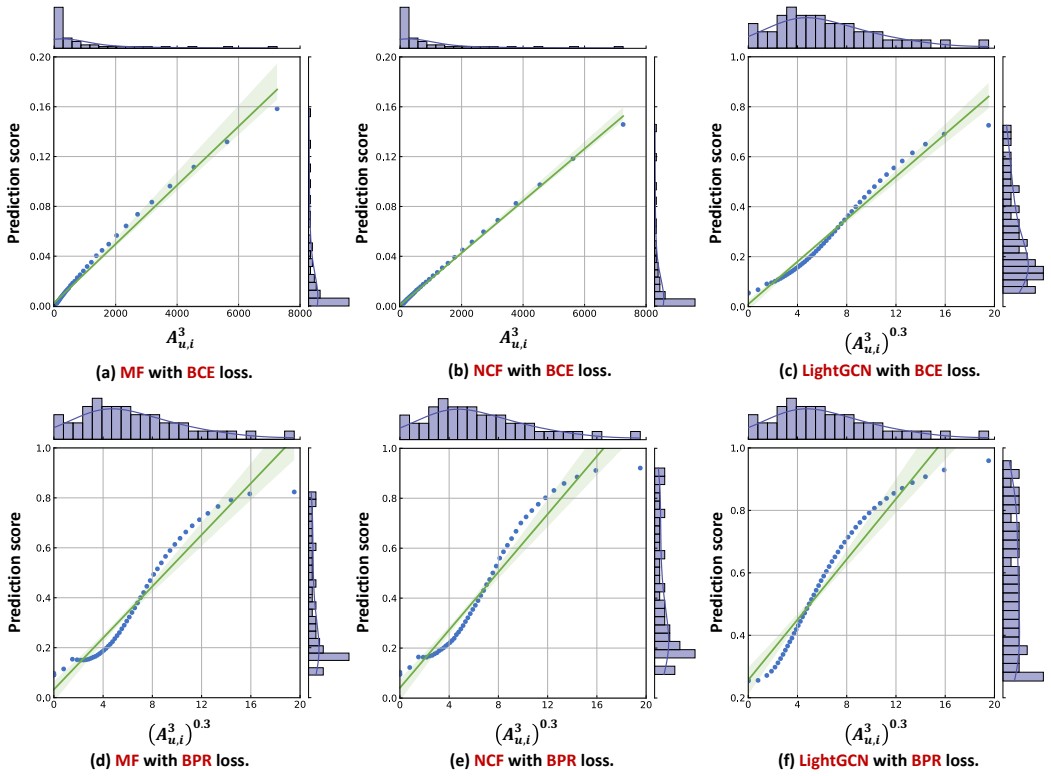

**Figure 7: Visualization of the correlation between the three-order path number and the normalized prediction scores from three representative CF models (*i.e.,* MF, NCF, LightGCN) with two widely used loss functions (*i.e.,* BCE and BPR). We conduct the Spearman Rank Correlation Test [63] for each figure. The test results with $r \geq 0.998 \approx 1$ and $p \leq 1.46e^{-59} \ll 0.001$ indicate the strong positive correlations in these experiments.**

where $\mathcal{N}_{i^*}$ denotes the first-order neighborhood set of item $i^*$ and $|\mathcal{N}_{u'}|$ denotes the size of $\mathcal{N}_{u'}$. Similarly, We utilize function $\phi_\theta(u, i, j)$ to represent $-[1 - \sigma(\mathcal{M}_\theta(u, i) - \mathcal{M}_\theta(u, j))]$ and assume its values on different $(u, i, j)$ pairs are approximately becoming closer as the training processes.

Based on the above derivation, we observe that the gradients of $e_{u^*}^\theta$ and $e_{i^*}^\theta$ always point towards a fixed target during the iteration, and their magnitudes are affected by $|\phi_\theta(u, i, j)|$, which gradually becomes smaller as the training converges. Therefore, we can approximate the converged values of $e_{u^*}^{\theta^*}$ and $e_{i^*}^{\theta^*}$ as:

$$e_{u^*}^{\theta^*} \approx C_1 \cdot \left[ \sum_{i \in \mathcal{N}_{u^*}} e_i^{\theta^*} - \frac{|\mathcal{N}_{u^*}|}{|\mathcal{I}|} \sum_{j \in \mathcal{I}} e_j^{\theta^*} \right], \tag{8}$$

$$e_{i^*}^{\theta^*} \approx C_1 \cdot \left[ \sum_{u \in \mathcal{N}_{i^*}} e_u^{\theta^*} - \frac{1}{|\mathcal{I}|} \sum_{u' \in \mathcal{U}} |\mathcal{N}_{u'}| \cdot e_{u'}^{\theta^*} \right], \tag{9}$$

where $C_1$ is a fixed constant. Furthermore, by using the average number of first-order neighbors $|\bar{\mathcal{N}}|$ of all nodes in the graph to approximate $|\mathcal{N}_{u'}|$ and $|\mathcal{N}_{i^*}|$, we can make the following assumption:

ASSUMPTION A.1. *When the model converges, the embeddings of user $u$ and item $i$ have the following form:*

$$e_{u^*}^{\theta^*} \approx C_1 \cdot \left[ \sum_{i \in \mathcal{N}_{u^*}} e_i^{\theta^*} - \frac{|\bar{\mathcal{N}}|}{|\mathcal{I}|} \sum_{j \in \mathcal{I}} e_j^{\theta^*} \right] \tag{10}$$

$$e_{i^*}^{\theta^*} \approx C_1 \cdot \left[ \sum_{u \in \mathcal{N}_{i^*}} e_u^{\theta^*} - \frac{|\bar{\mathcal{N}}|}{|\mathcal{I}|} \sum_{u' \in \mathcal{U}} e_{u'}^{\theta^*} \right], \tag{11}$$

*where $C_1$ is a fixed constant, $|\bar{\mathcal{N}}|$ denotes the average number of first-order neighbors of all nodes in the graph.*

Thus, we have proved the form of embeddings of users and items when the model converges. To complete the remaining proof, we need some additional assumptions.

ASSUMPTION A.2. *We assume that the collected user-item interactions for recommender training are sparse, which means that $\frac{|\bar{\mathcal{N}}|^3}{|\mathcal{I}|} \ll 1$.*

ASSUMPTION A.3. *When the model converges, the user-item pairs with historical interactions have larger prediction scores, while the user-item pairs without historical interactions have smaller predicted scores. This can be formulated as:*

$$(e_{u^*}^{\theta^*})^T (e_{i^*}^{\theta^*}) = \begin{cases} C_{large}, & if (u, i) \in \mathbf{P} \\ C_{small}, & if (u, i) \notin \mathbf{P} \end{cases}, \tag{12}$$

*where $\mathbf{P}$ denotes the positive interaction (positive user-item pair) set of $\mathbf{D}$ and $C_{small} \ll C_{large}$.*

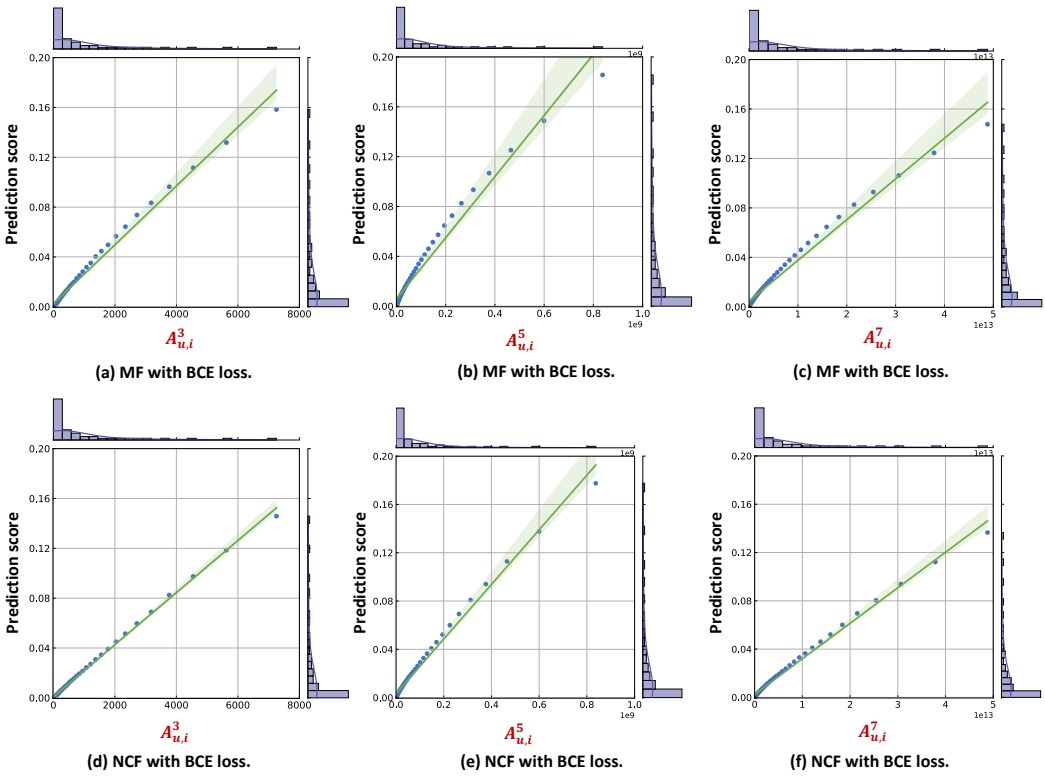

**Figure 8: Visualization of the correlation between the high-order path numbers and the prediction scores, where we vary the order number in $\{3, 5, 7\}$. The figures validate the robustness of such correlation in different order numbers. We conduct the Spearman Rank Correlation Test [63] for each figure. The test results with $r \geq 0.998 \approx 1$ and $p \leq 1.46e^{-59} \ll 0.001$ indicate the strong positive correlations in these experiments.**

The Assumptions A.2 and A.3 above are reasonable in recommender systems since the user interactions are usually sparse and neural CF models can well fit the training interactions. Hence, we could further derive the prediction score of a user-item pair when the model converges:

$$\mathcal{M}_{\theta^*}(u^*, i^*) = (e_{u^*}^{\theta^*})^T (e_{i^*}^{\theta^*})$$

$$\approx (C_1)^2 \cdot \left[ \sum_{i \in \mathcal{N}_{u^*}} e_i^{\theta^*} - \frac{|\bar{\mathcal{N}}|}{|\mathcal{I}|} \sum_{j \in \mathcal{I}} e_j^{\theta^*} \right]^T \left[ \sum_{u \in \mathcal{N}_{i^*}} e_u^{\theta^*} - \frac{|\bar{\mathcal{N}}|}{|\mathcal{I}|} \sum_{u' \in \mathcal{U}} e_{u'}^{\theta^*} \right]$$

$$\approx (C_1)^2 \cdot \left[ \sum_{i \in \mathcal{N}_{u^*}} \sum_{u \in \mathcal{N}_{i^*}} (e_u^{\theta^*})^T (e_i^{\theta^*}) - \frac{|\bar{\mathcal{N}}|}{|\mathcal{I}|} \sum_{u \in \mathcal{N}_{i^*}} \sum_{j \in \mathcal{I}} (e_j^{\theta^*})^T (e_u^{\theta^*}) \right.$$

$$\left. - \frac{|\bar{\mathcal{N}}|}{|\mathcal{I}|} \sum_{u' \in \mathcal{U}} \sum_{i \in \mathcal{N}_{u^*}} (e_{u'}^{\theta^*})^T (e_i^{\theta^*}) + \frac{|\bar{\mathcal{N}}|^2}{|\mathcal{I}|^2} \sum_{u' \in \mathcal{U}} \sum_{j \in \mathcal{I}} (e_{u'}^{\theta^*})^T (e_j^{\theta^*}) \right]. \tag{13}$$

Notably that

$$\sum_{i \in \mathcal{N}_{u^*}} \sum_{u \in \mathcal{N}_{i^*}} (e_u^{\theta^*})^T (e_i^{\theta^*})$$
$$= A_{u,i}^3 \cdot C_{\text{large}} + (|\mathcal{N}_{u^*}||\mathcal{N}_{i^*}| - A_{u,i}^3) \cdot C_{\text{small}} \tag{14}$$
$$\approx A_{u,i}^3 \cdot C_{\text{large}}.$$

Hence, by neglecting infinitesimal terms, the Eq. (13) can be reformulated as:

$$\mathcal{M}_{\theta^*}(u^*, i^*) \approx (C_1)^2 \cdot \left[ A_{u,i}^3 \cdot C_{\text{large}} - 2 \cdot \frac{|\bar{\mathcal{N}}|^2}{|\mathcal{I}|} \cdot C_{\text{large}} + \frac{|\bar{\mathcal{N}}|^3}{|\mathcal{I}|} \cdot C_{\text{large}} \right]$$

$$\approx (C_1)^2 \cdot A_{u,i}^3 \cdot C_{\text{large}}$$

$$\propto A_{u,i}^3. \tag{15}$$

So far, we have completed the proof of Eq. (4) in the special case with the MF model and BPR loss. Then we will generalize our conclusion to other CF models with different loss functions. To achieve it, we just need to prove that the assumptions still hold with other CF models. It is easy to observe that Assumption A.2 and Assumption A.3 remain valid. As to Assumption A.1, the derivation is also similar to NCF and LightGCN since the partial derivative of $\mathcal{L}_{\text{BPR}}$ with respect to the embeddings of users and items are similar. In the following, we prove that Assumption A.1 still holds for the CF models with BCE loss function. Formally, we have

$$\min_{\theta} \mathcal{L}_{\text{BCE}}(\mathcal{M}_{\theta}, D)$$

$$= \frac{1}{|P|} \sum_{(u,i) \in P} \left[ -\log(\sigma(\mathcal{M}_{\theta}(u, i))) - \frac{1}{|\mathcal{I}|} \sum_{j \in \mathcal{I}} \log(1 - \sigma(\mathcal{M}_{\theta}(u, j))) \right]. \tag{16}$$

Similarly, we could obtain the partial derivative of the loss function $\mathcal{L}_{\text{BCE}}$ with respect to the embeddings of user $u^*$ and item $i$ as Eq. (6)

---

**Algorithm 1** Dynamic Programming Algorithm

---

**Input:** a set of target users $\mathcal{U}_t$, treatment effects matrix $Y = \left[ Y^{\theta^*}_{u,i}(D_f(t_u)) \right]_{|\mathcal{U}_t| \times (H+1)}$ across different target users and budgets, the maximal budget number for each user $H$, and overall budget number $N$ for all target users.

1: **Function** dynamic_programming($|\mathcal{U}_t|, Y, H, N$)
2:  $B$ = 1D array of size $H + 1$ initialized with $[0, 1, 2, 3, ..., H]$;
3:      ▷ B contains the potential budget numbers of each user.
4:  $dp, selected$ = 2D array of size $(|\mathcal{U}_t| + 1) \times (N + 1)$ initialized with zeros;
5:    ▷ $selected$ is used to find the optimal allocation for each user.
6: **for** $i = 1$ to $|\mathcal{U}_t|$ **do**
7:    **for** $j = 0$ to $N$ **do**
8:      **for** $k = 0$ to $H$ **do**
9:        **if** $j \geq B[k]$ **then** then
10:          $dp[i][j] = \max(dp[i][j], dp[i-1][j - B[k]] + Y[i][k])$;
11:          $selected[i][j] = 1$;
12:        **end if**
13:      **end for**
14:    **end for**
15: **end for**
16: $P_{max} = dp[|\mathcal{U}_t|][N]$;
17: $T^*$ = 1D array of size $|\mathcal{U}_t| + 1$ initialized with zeros;
18: $j = N$;
19: **for** $i = |\mathcal{U}_t|$ down to 1 **do**
20:    $optimal\_budget = 0$;
21:    **for** $budget = 0$ to $H$ **do**
22:      **if** $j \geq B[budget]$ and $selected[i][j] = 1$ **then**
23:        $optimal\_budget = budget$;
24:        **Break**
25:      **end if**
26:      $T^*[i] \leftarrow optimal\_budget$;
27:      $j \leftarrow j - B[optimal\_budget]$
28:    **end for**
29: **end for**
30: **Return** $P_{max}, T^*$;

**Output:** overall recommendation probability $P_{max}$, and the optimal budget allocation $T^*$.

---

and Eq. (7):

$$
\begin{aligned}
\frac{\partial \mathcal{L}_{\text{BCE}}(\mathcal{M}_\theta, D)}{\partial e^\theta_{u^*}} &\approx \frac{1}{|P|} \sum_{i \in \mathcal{N}_{u^*}} \phi_\theta(u^*, i, j) \cdot e^\theta_i - \\
&\quad \frac{1}{|P|} \frac{|\mathcal{N}_{u^*}|}{|\mathcal{I}|} \sum_{j \in \mathcal{I}} \phi_\theta(u^*, i, j) \cdot e^\theta_j, \\
\frac{\partial \mathcal{L}_{\text{BCE}}(\mathcal{M}_\theta, D)}{\partial e^\theta_{i^*}} &\approx \frac{1}{|P|} \sum_{u \in \mathcal{N}_{i^*}} \phi_\theta(u, i^*, j) \cdot e^\theta_u - \\
&\quad \frac{1}{|P|} \frac{1}{|\mathcal{I}|} \sum_{u' \in \mathcal{U}} |\mathcal{N}_{u'}| \cdot \phi_\theta(u', i, i^*) \cdot e^\theta_{u'},
\end{aligned}
\tag{17}
$$

where $\phi_\theta(u, i, j) = -(1 - \sigma(\mathcal{M}_\theta(u, i)) = -\sigma(\mathcal{M}_\theta(u, j))$. Similarly, we can assume $\phi_\theta(u, i, j)$ approximately to become similar for different $(u, i, j)$ pairs and tends to be close to zero as training processes.

Therefore, we can conclude that Assumption A.1 also holds for the CF models (*e.g.*, MF, NCF, and LightGCN) with BCE loss. Due to the popularity of BCE and BPR losses in recommender optimization, we mainly study these two loss functions and leave the exploration of more loss functions to future work. □

### A.3 Theoretical Analysis of Proposition 2

**Reminder of Proposition 2.** The three-order path number between target user $u$ and target item $i$, $A^3_{u,i}$, is equivalent to the weighted sum of the intermediate users who liked item $i$ in $A$, where the weights are their interaction similarities with target user $u$, *i.e.*, the number of mutually liked items.

PROOF. If we denote all interactions of target item $i$ as $D^*_r \in \{0, 1\}^{|\mathcal{U}_r| \times 1}$, we can factorize $D_r$ into $\left[ \begin{array}{cc} \bar{D}_r & D^*_r \end{array} \right]$, where $\bar{D}_r \in \{0, 1\}^{|\mathcal{U}_r| \times (|\mathcal{I}| - 1)}$ represents the interactions on other items. In this way, the symmetric interaction matrix $A = \left[ \begin{array}{ccc} 0 & \bar{D}_r & D^*_r \\ (\bar{D}_r)^T & 0 & 0 \\ (D^*_r)^T & 0 & 0 \end{array} \right]$. Thereafter, the three-order path number from all users to the target item $i$ can be obtained from $A^3 = A \times A \times A$: $\bar{D}_r(\bar{D}_r)^T D^*_r + D^*_r(D^*_r)^T D^*_r$. Given a target user $u$, its three-order path number to the target item $i$ equals to a value in a row of $\bar{D}_r(\bar{D}_r)^T D^*_r + D^*_r(D^*_r)^T D^*_r$. Since there is no interaction between the target user $u$ and target item $i$ in $D^*_r$, their corresponding value in $D^*_r(D^*_r)^T D^*_r$ is also zero. As such, the three-order path number between $u$ and $i$ is only decided by $\bar{D}_r(\bar{D}_r)^T D^*_r$, where $\bar{D}_r(\bar{D}_r)^T$ describes the interaction similarity between all users (*i.e.*, the number of mutually liked items) and $D^*_r$ represents their preference over target item $i$. Afterward, the three-order path number from target user $u$ to target item $i$ is only affected by the users who liked item $i$ in $D^*_r$ while the sum is weighted by the interaction similarity with target user $u$ via $\bar{D}_r(\bar{D}_r)^T$. Therefore, we have completed the proof of Proposition 2.

□

### A.4 Dynamic Programming Algorithm

The budget-constrained treatment selection can be solved by using a dynamic programming approach for the knapsack problem, which is based on the principle of the optimal substructure. The dynamic programming algorithm efficiently computes the maximum value by iteratively solving smaller subproblems and integrating their solutions. Algorithm 1 presents pseudo-code for the process of calculating the optimal treatment after estimating the treatment effect of $Y^{\theta^*}_{u,i}(D_f(t_u))$. In this algorithm, the maximal budget for each user $H$ means that we can vary different budgets $t_u \in \{0, 1, 2, ..., H\}$ with $H \ll N$ to calculate $Y^{\theta^*}_{u,i}(D_f(t_u))$.

## B EXPERIMENTAL SETTINGS

### B.1 Hyper-parameter Tuning

In our UBA framework, we mainly introduce these hyper-parameters: fake user number $N$, the maximal fake user budget of each target user $H$, $\alpha$ and $\beta$ in $\alpha \cdot (A^3_{u,i})^\beta$, and the proportion of accessible user interactions. We mainly follow precious studies [28, 31, 32] for the selection of $H$ and $N$. As to $\alpha$ and $\beta$, they should be larger than zero to ensure positive correlations in Proposition 2. We choose

**Table 6: Evaluation on three datasets of all users when overall fake user budget is 100. ∗ implies the improvements over the best baseline "Target" are statistically significant ($p$-value<0.05) under $t$-test. Due to the hit ratios on ML-1M and Yelp being quite small, we multiply all the hit ratios of ML-1M and Yelp by 10 and 100, respectively, for better comparison.**

| | ML-1M | | | | Amazon | | | | Yelp | | | |
| | Popular item | | Unpopular item | | Popular item | | Unpopular item | | Popular item | | Unpopular item | |
| | HR@10 | HR@20 | HR@10 | HR@20 | HR@10 | HR@20 | HR@10 | HR@20 | HR@10 | HR@20 | HR@10 | HR@20 |
|---|---|---|---|---|---|---|---|---|---|---|---|---|
| Before Attack | 0.050 | 0.109 | 0.000 | 0.000 | 0.002 | 0.003 | 0.000 | 0.000 | 0.000 | 0.000 | 0.000 | 0.000 |
| Random Attack | 0.050 | 0.082 | 0.000 | 0.000 | 0.018 | 0.197 | 0.013 | 0.043 | 0.008 | 0.016 | 0.024 | 0.024 |
| Segment Attack | 0.069 | 0.123 | 0.008 | 0.012 | 0.014 | 0.145 | 0.034 | 0.044 | 0.008 | 0.016 | 0.032 | 0.064 |
| Bandwagon Attack | 0.059 | 0.119 | 0.000 | 0.002 | 0.015 | 0.176 | 0.025 | 0.063 | 0.032 | 0.064 | 0.008 | 0.016 |
| Average Attack | 0.016 | 0.044 | 0.002 | 0.002 | 0.016 | 0.185 | 0.013 | 0.048 | 0.016 | 0.048 | 0.040 | 0.056 |
| WGAN | 0.041 | 0.076 | 0.012 | 0.012 | 0.019 | 0.254 | 0.034 | 0.076 | 0.024 | 0.032 | 0.040 | 0.040 |
| DADA-DICT | 0.096 | 0.156 | 0.034 | 0.062 | 0.039 | 0.079 | 0.103 | 0.204 | 0.096 | 0.144 | 0.072 | 0.096 |
| DADA-DIV | 0.094 | 0.132 | 0.039 | 0.057 | 0.044 | 0.082 | 0.85 | 0.211 | 0.089 | 0.138 | 0.077 | 0.099 |
| DADA | 0.122 | 0.194 | 0.050 | 0.084 | 0.092 | 0.144 | 0.103 | 0.204 | 0.084 | 0.145 | 0.076 | 0.158 |
| AIA | 0.078 | 0.180 | 0.002 | 0.002 | 0.059 | 0.064 | 0.189 | 0.245 | 0.076 | 0.122 | 0.056 | 0.096 |
| + Target | 0.146 | 0.262 | 0.072 | 0.111 | 0.062 | 0.077 | 0.249 | 0.301 | 0.075 | 0.194 | 0.048 | 0.144 |
| + UBA(w/o $\mathcal{S}_\phi$) | 0.208 | 0.263 | 0.072 | 0.134 | 0.097 | 0.084 | 0.267 | 0.292 | 0.099 | 0.245 | 0.048 | 0.160 |
| + UBA(w/ $\mathcal{S}_\phi$) | 0.146 | 0.324 | 0.074 | 0.111 | 0.145 | 0.098 | 0.314 | 0.333 | 0.124 | 0.232 | 0.056 | 0.096 |
| AUSH | 0.082 | 0.183 | 0.000 | 0.002 | 0.103 | 0.138 | 0.064 | 0.098 | 0.056 | 0.111 | 0.024 | 0.049 |
| + Target | 0.146 | 0.260 | 0.062 | 0.082 | 0.176 | 0.378 | 0.082 | 0.095 | 0.048 | 0.120 | 0.056 | 0.168 |
| + UBA(w/o $\mathcal{S}_\phi$) | 0.154 | 0.296 | 0.076 | 0.106 | 0.189 | 0.395 | 0.082 | 0.121 | 0.096 | 0.128 | 0.064 | 0.152 |
| + UBA(w/ $\mathcal{S}_\phi$) | 0.190 | 0.371 | 0.067 | 0.092 | 0.189 | 0.423 | 0.071 | 0.189 | 0.072 | 0.128 | 0.064 | 0.184 |
| Leg-UP | 0.053 | 0.091 | 0.015 | 0.027 | 0.020 | 0.027 | 0.151 | 0.169 | 0.080 | 0.136 | 0.048 | 0.104 |
| + Target | 0.146 | 0.265 | 0.072 | 0.111 | 0.046 | 0.062 | 0.205 | 0.236 | 0.096 | 0.144 | 0.072 | 0.112 |
| + UBA(w/o $\mathcal{S}_\phi$) | 0.196 | 0.342 | 0.081 | 0.101 | 0.049 | 0.064 | 0.221 | 0.266 | 0.112 | 0.160 | 0.112 | 0.168 |
| + UBA(w/ $\mathcal{S}_\phi$) | 0.158 | 0.249 | 0.086 | 0.124 | 0.035 | 0.065 | 0.240 | 0.255 | 0.104 | 0.144 | 0.128 | 0.192 |

**Table 8: The table of time costs (minutes) on ML-1M.**

| Methods | AIA | AUSH | Leg-UP |
|---|---|---|---|
| One time estimation w/ $\mathcal{S}_\phi$ | 64.4m | 52.1m | 62.9m |
| Estimation w/o $\mathcal{S}_\phi$ | 1.1m | 1.1m | 1.1m |
| Whole attack process | 50.6m | 48.31m | 69.3m |

**Table 7: Performance of different backend models with UBA(w/o $S_\phi$) w.r.t. hyper-parameter tuning.**

| | Leg-UP | | AIA | | AUSH | |
| | HR@10 | HR@20 | HR@10 | HR@20 | HR@10 | HR@20 |
|---|---|---|---|---|---|---|
| $\alpha$= 1.0 $\beta$=1.0 | **0.18** | **0.32** | **0.24** | **0.4** | 0.28 | 0.40 |
| $\alpha$= 0.5 $\beta$=1.0 | 0.16 | 0.24 | 0.22 | 0.36 | **0.30** | 0.36 |
| $\alpha$= 1.0 $\beta$=0.3 | 0.18 | 0.30 | 0.24 | 0.38 | 0.26 | **0.42** |

$\alpha$ and $\beta$ from $\{0.5, 1\}$ and $\{0.3, 1\}$, respectively. As shown in Table 7, we find the UBA(w/o $\mathcal{S}_\phi$) is not very sensitive to $\alpha$ and $\beta$, and thus we choose $\alpha$ = 1.0, $\beta$ = 1.0 as our default parameters. Future work might consider more elaborate adjustments of $\alpha$ and $\beta$ in a larger scope. Lastly, we set the default proportion of user interactions accessible to attackers as 20% and provide more results with varying proportions in Table 3. As to the hyper-parameters of the backend attackers and recommender models, we follow their original settings.

## B.2 Resource Costs

All experiments are conducted on CentOS 1 machine with a 6-core Intel(R) Xeon(R) Gold 5218 CPU @ 2.30GHz, 1 NVIDIA GeForce RTX 3090 GPUs (24G), and 40G of RAM. In Table 8, we present the estimation time for the treatment effects, $Y_{u,i}^{\theta^*}(D_f(t_u))$, with and without using $\mathcal{S}_\phi$ on ML-1M. Additionally, we show the time of the whole attack process, from generating fake users by the attackers to obtaining recommendation results of the victim models. For each process, we conduct ten experiments and calculate the average time to ensure reliability.

For each estimation using the surrogate model $\mathcal{S}_\phi$, we can simultaneously obtain the recommendation probability of all target users under the same $t_u$, and the table shows the time used for such an estimation. Given limited overall fake user budgets $N$, the budgets a target user can obtain are limited ($H \ll N$). In our experiments, we usually use $H = 6$ for 100 limited fake user budgets on 50 target users, where $H = 6$ is sufficient to ensure a good attack performance. Since the estimation processes with different fake user budgets are independent, we can leverage multiple GPUs for parallel acceleration if necessary.

We can also observe that the estimation time costs without $\mathcal{S}_\phi$ are significantly lower. This is due to the high efficiency of calculating the three-order path numbers. By comparing the two estimation methods, we can find that: on one hand, using a reliable surrogate model results in more accurate effect estimation and better attack performance. However, training and attacking a surrogate model require more computational time during the estimation process. On the other hand, the estimation w/o $\mathcal{S}_\phi$ calculates the three-order path numbers in $A^3$, sacrificing little estimation accuracy while it is much faster in terms of computational time. In practical applications, these two estimation methods can be flexibly selected by the actual requirements.

## C MORE RESULTS

## C.1 Overall Attack Performance on All Users

In Table 6, we report the hit ratios on all users. Due to the different dataset sparsity, the hit ratios on the three datasets are not at the same magnitude. To better show the results, we multiply the hit ratios of ML-1M and Yelp by 10 and 100, respectively. Although the primary attack objective of target user attacks focuses on a specific set of users, UBA can also enhance the hit ratios of three backend attackers on all users, indicating that UBA not only achieves the SOTA attack performance on target users, but also guarantees the

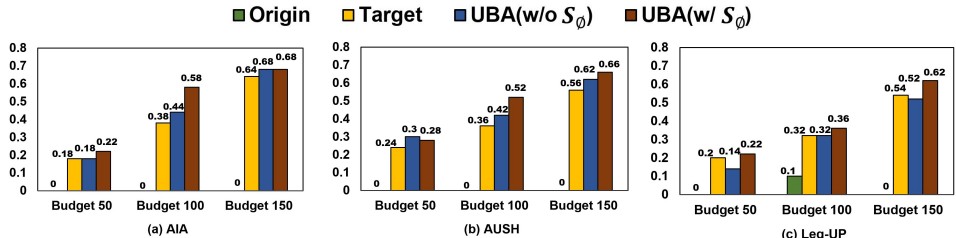

**Figure 9: Performance comparison *w.r.t.* HR@10 under different attack budgets.**

**Table 9: Attack performance comparison w/ and w/o using defense models on all users.**

| | | AIA | | + Target | | + UBA(w/o $\mathcal{S}_\phi$) | | + UBA(w/ $\mathcal{S}_\phi$) | |
| | | Origin | Detector | Origin | Detector | Origin | Detector | Origin | Detector |
|---|---|---|---|---|---|---|---|---|---|
| PCA | HR@10 | 0.0078 | 0.0087 | 0.0146 | 0.0132 | 0.0208 | 0.0182 | 0.0146 | 0.0141 |
| | HR@20 | 0.0180 | 0.0192 | 0.0262 | 0.0251 | 0.0324 | 0.0235 | 0.0263 | 0.0240 |
| FAP | HR@10 | 0.0078 | 0.0087 | 0.0146 | 0.0132 | 0.0208 | 0.0182 | 0.0146 | 0.0141 |
| | HR@20 | 0.0180 | 0.0192 | 0.0262 | 0.0251 | 0.0324 | 0.0235 | 0.0263 | 0.0240 |
| | | **AUSH** | | **+ Target** | | **+ UBA(w/o $\mathcal{S}_\phi$)** | | **+ UBA(w/ $\mathcal{S}_\phi$)** | |
| PCA | HR@10 | 0.0082 | 0.0107 | 0.0146 | 0.0142 | 0.0154 | 0.0149 | 0.0168 | 0.0210 |
| | HR@20 | 0.0183 | 0.0203 | 0.0260 | 0.0260 | 0.0296 | 0.0281 | 0.0287 | 0.0336 |
| FAP | HR@10 | 0.0082 | 0.0068 | 0.0146 | 0.0130 | 0.0154 | 0.0084 | 0.0168 | 0.0093 |
| | HR@20 | 0.0183 | 0.0125 | 0.0260 | 0.0228 | 0.0296 | 0.0172 | 0.0287 | 0.0169 |
| | | **Leg-UP** | | **+ Target** | | **+ UBA(w/o $\mathcal{S}_\phi$)** | | **+ UBA(w/ $\mathcal{S}_\phi$)** | |
| PCA | HR@10 | 0.0053 | 0.0025 | 0.0146 | 0.0091 | 0.0196 | 0.0146 | 0.0158 | 0.0160 |
| | HR@20 | 0.0091 | 0.0044 | 0.0265 | 0.0181 | 0.0342 | 0.0258 | 0.0249 | 0.0263 |
| FAP | HR@10 | 0.0053 | 0.0025 | 0.0146 | 0.0089 | 0.0196 | 0.0091 | 0.0158 | 0.0130 |
| | HR@20 | 0.0091 | 0.0044 | 0.0265 | 0.0160 | 0.0342 | 0.0254 | 0.0249 | 0.0228 |

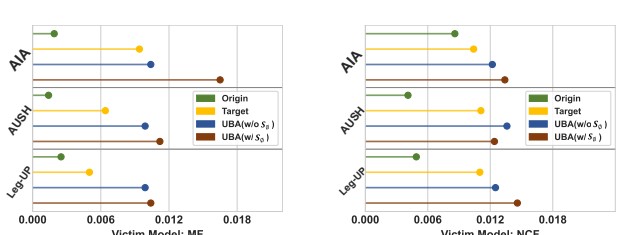

**Figure 10: Generalization of UBA *w.r.t.* HR@10 across different victim models on all users.**

attack results on all users in the platform. In this light, UBA can also be utilized for injective attacks on all users, improving the attack performance of existing injective attackers.

## C.2 Attack Performance on All Users across Victim Models

Figure 10 visualizes the attack results of using MF and NCF as victim models on ML-1M. By inspecting Figure 10 and the results of LightGCN in Table 6, we can have the observations that UBA shows higher hit ratios than "Origin" and "Target" on all users, indicating good generalization ability of UBA across different victim models even for all users.

## C.3 Attack Performance on All Users with Defense Models

When defense models are deployed, we present the attack results of three backend attackers with "Target" and UBA on all users of ML-1M in Table 9. From the table, we find 1) the detectors are still effective in most cases although they are not quite robust, and 2)

UBA methods can still have superior attack performance than the baselines when the detectors are applied.

## C.4 Attack of Unpopular Items with Varying Budgets

In Figure 9, we present the attack performance with the unpopular target items in ML-1M *w.r.t.* varying fake user budgets. We can find that the trends on the unpopular items are similar to those on the popular items presented in Figure 3. For popular or unpopular items, both UBA w/ and w/o $\mathcal{S}_\phi$ achieve better attack performance than the backend attackers and "Target" under different budgets. This further verifies the robustness of UBA *w.r.t.* different fake user budgets and the target items with different popularity.

## D RELATED WORK

### D.1 Uplift Modeling

Uplift, a term commonly used in marketing, refers to the disparity in purchasing behaviors between customers who receive a promotional offer (the treated group) and those who do not (the control group) [35, 68]. In causal terms, uplift essentially measures the causal effect of a treatment, such as a promotion, on the desired outcome, such as customer purchasing behaviors. While uplift modeling has been extensively researched in the fields of machine learning and marketing[1, 18, 46], its application in the realm of recommendation systems has received relatively little attention [52, 54, 59].

Initial studies in this area have primarily focused on exploring the potential of uplift modeling to regulate the exposure proportion of different item categories [62]. However, in this work, we take a different perspective by defining the assigning of fake user budgets in injective attacks as the treatment variable. We estimate the difference in recommendation probabilities of the target item on target users as the uplifts. Leveraging these estimated uplifts, our objective is to identify the optimal treatment strategy that maximizes the overall recommendation probabilities for all target users.

Target user attacks usually have limited fake user budgets, which can be formulated as a budget-constrained optimization problem. Therefore, the core is how to maximize the attack performance by using limited budgets. By adopting uplift modeling techniques in the context of recommender attacks, we can optimally utilize the limited budgets to increase their causal effects.

### D.2 Injective Attacks

Injective attacks aim to disrupt the recommendation strategy of a victim recommender model in order to increase the exposure of a

target item to all users [11, 37, 38, 60]. To achieve this, attackers often inject fake users into the training data of the victim model for interference. As such, the core of injective attacks is to construct the interaction behaviors of fake users.

The construction methods of fake user interactions can be mainly categorized into three types:

- Heuristic attackers create fake users based on some heuristic rules. The goal is to enhance the similarity between fake users and real users while improving the co-occurrence of the target item and some selected items, thus increasing the exposure probability of the target item. Existing methods, such as Random Attack, Average Attack, Bandwagon Attack, and Segment Attack, usually leverage different human-designed rules to determine the selected items [6, 7, 24, 33].
- Gradient attacks optimize the attack objective in a continuous space to directly adjust the fake user interactions, and then truncate continuous values into discrete fake user interactions [15, 16, 23, 27, 58, 64].
- Neural attacks leverage generative neural networks to generate fake user interactions. To maximize the attack objective, various methods, such as WGAN, AIA, AUSH, and Leg-UP, have been proposed to improve the effectiveness of neural attackers [3, 31, 32, 44]. Besides, reinforcement learning is also employed when the attacker can rely on some sparse feedback from the victim model [14, 43, 69].

Data security and privacy concerns are making the assumption of attack knowledge increasingly important. Attack knowledge assumptions can be categorized into white-box, grey-box, and black-box settings. White-box settings assume that the structure and parameters of the victim recommender model, as well as all user-item interaction data, are accessible [15, 16, 27]. Besides, grey-box settings have limited access to the user-item interactions and knowledge about the victim model [9, 50, 58]. Lastly, black-box settings can only have a few feedback from the spy users [7]. In this work, we utilize a reasonable setting, where only the interactions of partial users are accessible to the attackers and the victim models are totally unknown.

## D.3 Defense Methods

A defense model plays a crucial role in preventing fake users from disturbing victim models. Existing methods can be roughly classified into two categories. One direction is to design more robust recommender models against injective attacks [49, 51]. However, this line of methods cannot be applied to current well-designed industry recommender models. To complement this, some studies focus on detecting and excluding fake users from the training data of recommender models [2, 5].

Based on whether the detection methods use the labels of fake users for training, current direction methods can be categorized into three types: supervised models [13, 30, 57], semi-supervised models [8, 53], and unsupervised models [10, 36, 66, 71]. Technically speaking, these methods have explored sequential GANs [41], RNNs [17], and CNNs [61] for detection. In future work, these methods can be evaluated by comparing the performance in defending against target user attackers.

In addition, a more critical direction is adversarial training [21, 34, 45]. The defense models can be trained with some transparent attackers via multi-turn attack and defense, enhancing the robustness and generalization of defense models in the presence of fake user data. Our UBA framework can be applied to improve extensive attackers, serving as a great assistant for such adversarial training.