# OpenReview forum: "Uplift Modeling for Target User Attacks on Recommender Systems"
_ACM.org/TheWebConf/2024/Conference — TheWebConf24 Oral_

### Official Review · Reviewer_sZHj · 2023-11-22

**Novelty:** 5
**Technical Quality:** 6

**Review:**

Summary
This paper investigates target user attacks against recommender systems. The authors propose to migrate the methodology of traditional injective attacks to target user attacks by assigning a budget to each target user. They propose a model-agnostic approach, UBA, to get the optimal budget allocation plan. To estimate the effect of different budget allocation plans, they propose two estimation methods, namely w/ and w/o. w/ uses the change in the output of the surrogate model after the attack to estimate the effect of the budget allocation plan. w/o uses the number of high-order interaction paths between the user and the item to estimate the effect of the budget allocation plan. Finally, they experimentally validate the effectiveness of the method.

Pros
1. The authors propose a model-agnostic approach to migrate the methodology of traditional injective attacks to target user attacks, which is interesting and effective.
2. The authors find a correlation between recommendation prediction scores and the number of high-order interaction paths, which is simple but interesting.
3. The paper is well-structured.


Cons
1. The authors estimate the treatment effect for each target user individually, but in practice the attack on one user is likely to result in a change in the recommendation for other users. Therefore, there are limitations in assigning budgets to each target user and decomposing multi-targeted target user attacks into multiple single-targeted attacks.
2. The authors used existing attack methods directly for experiments.  It would be better to propose new attack algorithms based on the UBA architecture.
3. In the hyper-parameter tuning section, the choices of α and β are too few.

**Questions:**

1. Proposition 1 is derived on which dataset? Has it been validated on other datasets?
2.In Section 4.1, the authors present two hyperparameters, α and β, and assume that they are invariant after the attack. I would like to know if there is any reason for this.
3. Since Section 4.1 yields a strong correlation between recommendation prediction scores and number of high-order interaction paths, why not generate attacks based on it directly? For example, according to Proposition 2, we can copy the target user's interactions as the fake user's interactions and add an interaction with target item, which maximizes the number of three-order paths.
4. Why have Target and UBA groups only experimented on AIA, AUSH, Legup?
5. In Algorithm 1, B[k] is always equal to k. What is the significance of B?
6. Why are "Target" and "UBA" groups only experimented on AIA, AUSH and Legup?
7.Why does the "Target" group in Table 6 outperform the original attack algorithm despite having an optimization objective different from the attack objective?

**Reviewer Confidence:**

3: The reviewer is confident but not certain that the evaluation is correct

**Scope:**

4: The work is relevant to the Web and to the track, and is of broad interest to the community

---

### Official Review · Reviewer_NQFn · 2023-11-23

**Novelty:** 5
**Technical Quality:** 5

**Review:**

The paper focuses on target user attacks in recommender systems. These attacks aim to manipulate the exposure of specific items to a particular user group. The key novelty is the introduction of the Uplift-guided Budget Allocation (UBA) framework, which optimizes the allocation of fake user budgets based on the estimated treatment effect on each target user. This approach aims to maximize attack performance while addressing the varying difficulty of attacking different users. The framework was empirically tested on three datasets under various scenarios, including different target items, user groups, budget constraints, victim models, and defense models, to validate its effectiveness and robustness.

Pros:
1.	Similar to some existing work, the proposed approach (UBA) optimizes the fake user interaction matrix. In addition, UBA considers varying attack difficulty of each user in optimization. By optimizing the allocation of fake user budgets through the UBA framework, the paper presents a more resource-efficient approach to carrying out attacks, maximizing the impact with minimal resources.
2.	The framework is tested across multiple scenarios and datasets, which provides a validation of its effectiveness and robustness.

Cons:
1.	The main contribution is target user attack with varying user attack difficulty.  As mentioned in the paper, to keep the optimization simple, the paper adopts two disjoint steps including the estimation of the treatment effect Y(u,i)Df and the selection of treatment. The treatment effect can be estimated either through surrogate model and simulations or high-order interaction path (A3). In addition, the proposed method computes the optimal budget based on a dynamic programming algorithm. Those two are not jointly optimized and some steps are computational expensive.

2.	Among the two estimation methods, UBA with surrogate models seems more computational expensive. Is there any quantitively analysis of the accuracy-efficiency trade-off for UBA/with surrogate and UBA/with A3?

3.	Only simple unsupervised defending methods (PCA, FAP) are discussed in the last section. Adding more recent defending methods such as “Denoise unreliable interactions for graph collaborative filtering” would have added more value to the paper.

**Questions:**

1.	Neural collaborative filtering (neurl CF) models are widely used models and perform better than matrix factorization (MF). For A3, is it easy to extend the proof from matrix factorization models to neural models?

**Reviewer Confidence:**

3: The reviewer is confident but not certain that the evaluation is correct

**Scope:**

4: The work is relevant to the Web and to the track, and is of broad interest to the community

---

### Official Review · Reviewer_N8KH · 2023-11-24

**Novelty:** 4
**Technical Quality:** 4

**Review:**

Paper summary:

This paper proposes an approach to addressing the vulnerability of recommender systems to injective attacks. By focusing on target user attacks and formulating varying attack difficulty as heterogeneous treatment effects, the Uplift-guided Budget Allocation (UBA) framework optimizes the allocation of fake user budgets to maximize attack performance. The paper presents theoretical and empirical analysis to demonstrate the rationality and effectiveness of UBA and validates its robustness against defense models through extensive experiments on three datasets under various settings. The paper also highlights the significance of target user attacks and introduces related literature on uplift modeling and injective attacks.

===

Pros:

P1. The paper investigates injective attacks on recommender systems, which can inspire existing industries to resist attacks.

P2. The paper validates the effectiveness of the proposed approach through extensive experiments on three datasets under various settings.

P3. The paper provides a comprehensive review of related literature on uplift modeling and injective attacks, which helps readers understand the context and significance of the proposed approach.

===

Cons:

C1. The datasets for evaluation seem to be limited.

C2. The efficiency of the proposed approach needs to be investigated since the budget allocation problem is NP-hard

C3. The writing and experiment settings can be improved.

**Questions:**

Q1. The paper uses only three small datasets to evaluate the proposed approach, which may not be representative of all possible scenarios. It would be useful to test the approach on more large datasets to validate its generalizability.

Q2. The efficiency of the proposed framework is not well investigated. It is better to add the time cost with different budgets and target users since the budget allocation problem is NP-hard regarding the number of budgets and target users.


Q3. The comparisons with baselines seem to be unfair. The baselines consider all users for promotion, while this paper only focuses on ~100 target users. However, the hit ratio is only computed based on these target users, so it is better to simply extend baselines in the same scenarios.

Q4. Some parts are unclear. For example, in Table 8, why the time of the whole attack process is longer than one-time estimations?

**Reviewer Confidence:**

2: The reviewer is willing to defend the evaluation, but it is likely that the reviewer did not understand parts of the paper

**Scope:**

2: The connection to the Web is incidental, e.g., use of Web data or API

---

### Official Review · Reviewer_ufRF · 2023-11-25

**Novelty:** 5
**Technical Quality:** 6

**Review:**

The paper proposes a novel attack approach against recommender systems called Uplift-guided Budget Allocation (i.e., UBA) which calculates the optimal allocation of budgets to generate and inject fake users into the system. Indeed, the initial authors’ assumption is that not all users might be interested in a specific target item for the attack (as users are usually clustered according to their expressed preferences). Thus, to not waste the budget for users who are unwilling to interact with the target attack item, the authors decide to produce attacks that are specifically tailored for each user. As the UBA framework is model-agnostic and it is assumed that the actual recommendation system is not known in advance (i.e., black-box scenario) the authors propose two variants where the attack leverages a surrogate recommendation model for the real one (e.g., MF) or tries to simulate the user’s path within the user-item graph in a random walk-alike manner with three walks. The UBA framework is tested in a suite of 10 state-of-the-art other attack strategies, and three possible recommendation systems as surrogate models, on three recommendation datasets. Experiments, which are further conducted on an extensive set of evaluation dimensions, demonstrate the efficacy of the proposed approach in all its components and design strategies.
*Pros*:
- The authors extensively state the current issues and how their framework may address them
- The paper is well-structured and easy to follow
- The proposed methodology is sound thanks to the theoretical and empirical demonstrations/intuitions provided by the authors in the main and appendix parts of the paper
- The experimental setting is extensive with several baselines, recommendation datasets, and evaluation dimensions
*Cons*:
- Despite the code being shared at review time, the provided URL seems to be broken

*Detailed comments*
Overall, the paper proposes a very nice approach well-placed in the existing literature, outlining the critical aspects of the current solutions and how the introduced framework may address them. From a structural viewpoint, the paper is well-written, and its narrative is easy to follow even for those who are not very familiar with the main topics of the work. In terms of methodology, the UBA framework seems to be adequately sound in all its formulations and theoretical foundations, which are extensively investigated and demonstrated (especially in the appendix). Finally, the experimental setting is extensive as several baselines are tested, on a sufficient group of recommendation datasets, and numerous evaluation dimensions are considered to encompass all possible facets of the approach. The only negative aspect I can see here is that the code, despite being shared, is not accessible at review time (maybe the URL is broken or expired).

**Questions:**

- Can the authors provide intuitions or mathematical proofs of how the UBA framework may work better in the setting without the surrogate model?
- To the best of my knowledge, I see that the intuition (theoretically and empirically demonstrated in the appendix) that the three-order path $A^3$ is positively correlated to the prediction score of any CF model is something not completely new to the field. Indeed, also the authors from [*] used the three-hop adjacency matrix to simulate the path of the user exploring distant items. In light of this, can the authors explain which are the connections between their intuitions/demonstrations and the cited paper?

[*] Bibek Paudel, Fabian Christoffel, Chris Newell, Abraham Bernstein: Updatable, Accurate, Diverse, and Scalable Recommendations for Interactive Applications. ACM Trans. Interact. Intell. Syst. 7(1): 1:1-1:34 (2017)

**Ethics Review Description:**

No issue

**Reviewer Confidence:**

3: The reviewer is confident but not certain that the evaluation is correct

**Scope:**

4: The work is relevant to the Web and to the track, and is of broad interest to the community

---

### Official Review · Reviewer_JrHP · 2023-11-26

**Novelty:** 5
**Technical Quality:** 6

**Review:**

### Summary

This paper presents a framework for a new injective attacks of recommender systems, which is the target user attacks. The authors proposed a new framework called Uplift-guided Budget Allocation (UBA) and show the attacker effectiveness on three datasets with various settings.

### Strength

1. Is it an interesting and meaningful setting to attack target user groups, i.e., expose a target item to a specific user group instead of all users.
2. This model is technically sound.
3. The effectiveness of this proposed method has been justified with extensive experiments.

### Weakness

1. Related work: recently a group of new injective recommender attacks paper based on model extraction [1-3], which is missed in the introduction about the existing attacks.
2. The authors provided an expired repo link, so the reproducibility cannot be verified.

### Reference

1. Yue, Zhenrui, et al. "Black-box attacks on sequential recommenders via data-free model extraction." *Proceedings of the 15th ACM Conference on Recommender Systems*. 2021.
2. Chen, Jingfan, et al. "Knowledge-enhanced black-box attacks for recommendations." *Proceedings of the 28th ACM SIGKDD Conference on Knowledge Discovery and Data Mining*. 2022.
3. Nguyen, Thanh Toan, et al. "Poisoning GNN-based recommender systems with generative surrogate-based attacks." *ACM Transactions on Information Systems* (2022).

**Questions:**

1. Is there any discussion or consideration of ethical aspects related to the attacking methods?

**Reviewer Confidence:**

2: The reviewer is willing to defend the evaluation, but it is likely that the reviewer did not understand parts of the paper

**Scope:**

3: The work is somewhat relevant to the Web and to the track, and is of narrow interest to a sub-community

---

### Decision · Program_Chairs · 2024-01-22

**Decision:**

Accept (Oral)

**Comment:**

The reviewers were broadly very positive about this work, both in terms of novelty and technical quality. The authors provided in-depth explanations and more details in the discussion phase. Code and data are shared by the authors for reproducibility. Overall, this could be a solid contribution to the conference.